# List-decodeable Linear Regression

**Sushrut Karmalkar**
University of Texas at Austin
sushrutk@cs.utexas.edu

**Adam R. Klivans**
University of Texas at Austin
klivans@cs.utexas.edu

**Pravesh K. Kothari**
Princeton University and Institute for Advanced Study
kothari@cs.princeton.edu

## Abstract

We give the first polynomial-time algorithm for robust regression in the list-decodable setting where an adversary can corrupt a greater than $1/2$ fraction of examples.

For any $\alpha < 1$, our algorithm takes as input a sample $\{(x_i, y_i)\}_{i \leq n}$ of $n$ linear equations where $\alpha n$ of the equations satisfy $y_i = \langle x_i, \ell^* \rangle + \zeta$ for some small noise $\zeta$ and $(1 - \alpha)n$ of the equations are *arbitrarily* chosen. It outputs a list $L$ of size $O(1/\alpha)$ - a fixed constant - that contains an $\ell$ that is close to $\ell^*$.

Our algorithm succeeds whenever the inliers are chosen from a *certifiably* anti-concentrated distribution $D$. As a corollary of our algorithmic result, we obtain a $(d/\alpha)^{O(1/\alpha^8)}$ time algorithm to find a $O(1/\alpha)$ size list when the inlier distribution is standard Gaussian. For discrete product distributions that are anti-concentrated only in *regular* directions, we give an algorithm that achieves similar guarantee under the promise that $\ell^*$ has all coordinates of the same magnitude. To complement our result, we prove that the anti-concentration assumption on the inliers is information-theoretically necessary.

To solve the problem we introduce a new framework for list-decodable learning that strengthens the "identifiability to algorithms" paradigm based on the sum-of-squares method.

## 1  Introduction

In this work, we design algorithms for the problem of linear regression that are robust to training sets with an overwhelming ($\gg 1/2$) fraction of adversarially chosen outliers.

Outlier-robust learning algorithms have been extensively studied (under the name *robust statistics*) in mathematical statistics [43, 37, 25, 23]. However, the algorithms resulting from this line of work usually run in time exponential in the dimension of the data [6]. An influential line of recent work [29, 1, 16, 33, 8, 30, 31, 24, 14, 17, 28] has focused on designing *efficient* algorithms for outlier-robust learning.

Our work extends this line of research. Our algorithms work in the "list-decodable learning" framework. In this model, a majority of the training data (a $1 - \alpha$ fraction) can be adversarially corrupted leaving only an $\alpha \ll 1/2$ fraction of "inliers". Since uniquely recovering the underlying parameters is information-theoretically *impossible* in such a setting, the goal is to output a list (with an absolute constant size) of parameters, one of which matches the ground truth. This model was introduced in [3] to give a discriminative framework for clustering. More recently, beginning with [8], various works [18, 30] have considered this as a model of "untrusted" data.

There has been phenomenal progress in developing techniques for outlier-robust learning with a *small* ($\ll 1/2$)-fraction of outliers (e.g. outlier "filters" [13, 14, 10, 15], separation oracles for inliers [13] or the *sum-of-squares* method [31, 24, 30, 28]). In contrast, progress on algorithms that tolerate the significantly harsher conditions in the list-decodable setting has been slower. The only prior works [8, 18, 30] in this direction designed list-decodable algorithms for mean estimation via problem-specific methods. Recently [22] addressed the somewhat related problem of conditional linear regression where the goal is to find a linear function with small square loss *conditioned* on a subset of training points whose 'indices' satisfy some constant-width $k$-DNF formula.

In this paper, we develop a principled technique to give the first efficient list-decodable learning algorithm for the fundamental problem of *linear regression*. Our algorithm takes a corrupted set of linear equations with an $\alpha \ll 1/2$ fraction of inliers and outputs a $O(1/\alpha)$-size list of linear functions, one of which is guaranteed to be close to the ground truth (i.e., the linear function that correctly labels the inliers). A key conceptual insight in this result is that list-decodable regression information-theoretically requires the inlier-distribution to be "anti-concentrated". Our algorithm succeeds whenever the distribution satisfies a stronger "certifiable anti-concentration" condition that is algorithmically "usable'. This class includes the standard gaussian distribution and more generally, any spherically symmetric distribution with strictly sub-exponential tails.

Prior to our work[1], the state-of-the-art outlier-robust algorithms for linear regression [28, 19, 12, 39] could handle only a small ($< 0.1$)-fraction of outliers even under strong assumptions on the underlying distributions.

List-decodable regression generalizes the well-studied [11, 26, 21, 44, 2, 9, 45, 41, 34] and *easier* problem of *mixed linear regression*: given $k$ "clusters" of examples that are labeled by one out of $k$ distinct unknown linear functions, find the unknown set of linear functions. All known techniques for the problem rely on faithfully estimating certain *moment tensors* from samples and thus, cannot tolerate the overwhelming fraction of outliers in the list-decodable setting. On the other hand, since we can take any cluster as inliers and treat rest as outliers, our algorithm immediately yields new efficient algorithms for mixed linear regression. Unlike all prior works, our algorithms work without any pairwise separation or bounded condition-number assumptions on the $k$ linear functions.

**List-Decodable Learning via the Sum-of-Squares Method**   Our algorithm relies on a strengthening of the robust-estimation framework based on the sum-of-squares (SoS) method. This paradigm has been recently used for clustering mixture models [24, 30] and obtaining algorithms for moment estimation [31] and linear regression [28] that are resilient to a small ($\ll 1/2$) fraction of outliers under the mildest known assumptions on the underlying distributions. At the heart of this technique is a reduction of outlier-robust algorithm design to just finding "simple" proofs of unique "identifiability" of the unknown parameter of the original distribution from a corrupted sample. However, this principled method works only in the setting with a small ($\ll 1/2$) fraction of outliers. As a consequence, the work of [30] for mean estimation in the list-decodable setting relied on "supplementing" the SoS method with a somewhat *ad hoc*, problem-dependent technique.

As an important conceptual contribution, our work yields a framework for list-decodable learning that recovers some of the simplicity of the general blueprint. Central to our framework is a general method of *rounding by votes* for "pseudo-distributions" in the setting with $\gg 1/2$ fraction outliers. Our rounding builds on the work of [32] who developed such a method to give a simpler proof of the list-decodable mean estimation result of [30]. In Section 2, we explain our ideas in detail.

The results in all the works above hold for any underlying distribution that has upper-bounded low-degree moments and such bounds are "captured" within the SoS system. Such conditions are called as "certified bounded moment" inequalities. An important contribution of this work is to formalize *anti-concentration* inequalities within the SoS system and prove "certified anti-concentration" for natural distribution families. Unlike bounded moment inequalities, there is no canonical encoding within SoS for such statements. We choose an encoding that allow proving certified anti-concentration for a distribution by showing the existence of a certain approximating polynomial. This allows showing certified anti-concentration of natural distributions via a completely modular approach that relies on a beautiful line of works that construct "weighted " polynomial approximators [35].

We believe that our framework for list-decodable estimation and our formulation of certified anti-concentration condition will likely have further applications in outlier-robust learning.

## 1.1 Our Results

We first define our model for generating samples for list-decodable regression.

**Model 1.1** (Robust Linear Regression). For $0 < \alpha < 1$ and $\ell^* \in \mathbb{R}^d$ with $\|\ell^*\|_2 \leq 1$, let $\mathrm{Lin}_D(\alpha, \ell^*)$ denote the following probabilistic process to generate $n$ noisy linear equations $\mathcal{S} = \{\langle x_i, a \rangle = y_i \mid 1 \leq i \leq n\}$ in variable $a \in \mathbb{R}^d$ with $\alpha n$ *inliers* $\mathcal{I}$ and $(1 - \alpha)n$ *outliers* $\mathcal{O}$:

1. Construct $\mathcal{I}$ by choosing $\alpha n$ i.i.d. samples $x_i \sim D$ and set $y_i = \langle x_i, \ell^* \rangle + \zeta$ for additive noise $\zeta$,

2. Construct $\mathcal{O}$ by choosing the remaining $(1 - \alpha)n$ equations arbitrarily and potentially adversarially w.r.t the inliers $\mathcal{I}$.

Note that $\alpha$ measures the "signal" (fraction of inliers) and can be $\ll 1/2$. The bound on the norm of $\ell^*$ is without any loss of generality. For the sake of exposition, we will restrict to $\zeta = 0$ for most of this paper and discuss (see Remarks 1.6 and 4.4) how our algorithms can tolerate additive noise.

An $\eta$-approximate algorithm for list-decodable regression takes input a sample from $\mathrm{Lin}_D(\alpha, \ell^*)$ and outputs a *constant* (depending only on $\alpha$) size list $L$ of linear functions such that there is some $\ell \in L$ that is $\eta$-close to $\ell^*$.

One of our key conceptual contributions is to identify the strong relationship between *anti-concentration inequalities* and list-decodable regression. Anti-concentration inequalities are well-studied [20, 42, 40] in probability theory and combinatorics. The simplest of these inequalities upper bound the probability that a high-dimensional random variable has zero projections in any direction.

**Definition 1.2** (Anti-Concentration). A $\mathbb{R}^d$-valued zero-mean random variable $Y$ has a $\delta$-*anti-concentrated* distribution if $\Pr[\langle Y, v \rangle = 0] < \delta$.

In Proposition 2.4, we provide a simple but conceptually illuminating proof that anti-concentration is *sufficient* for list-decodable regression. In Theorem 6.1, we prove a sharp converse and show that anti-concentration is information-theoretically *necessary* for even noiseless list-decodable regression. This lower bound surprisingly holds for a natural distribution: uniform distribution on $\{0, 1\}^d$ and more generally, uniform distribution on $[q]^d$ for $q = \{0, 1, 2 \ldots, q\}$. And in fact, our lower bound shows the impossibility of even the "easier" problem of mixed linear regression on this distribution.

**Theorem 1.3** (See Proposition 2.4 and Theorem 6.1). *There is a (inefficient) list-decodable regression algorithm for $\mathrm{Lin}_D(\alpha, \ell^*)$ with list size $O(\frac{1}{\alpha})$ whenever $D$ is $\alpha$-anti-concentrated. Further, there exists a distribution $D$ on $\mathbb{R}^d$ that is $(\alpha + \epsilon)$-anti-concentrated for every $\epsilon > 0$ but there is no algorithm for $\frac{\alpha}{2}$-approximate list-decodable regression for $\mathrm{Lin}_D(\alpha, \ell^*)$ that returns a list of size $< d$.*

To handle additive noise of variance $\zeta^2$, we need a control of $\Pr[|\langle x, v \rangle| \leq \zeta]$. For our efficient algorithms, in addition, we need a *certified* version of the anti-concentration condition. Informally, this means that there is a "low-degree sum-of-squares proof" of anti-concentration of $\mathcal{I}$. We give precise definition and background in Section 3. For this section, we will use this phrase informally and encourage the reader to think of it as a version of anti-concentration that the SoS method can reason about.

**Definition 1.4** (Certifiable Anti-Concentration). A random variable $Y$ has a $k$-*certifiably* $(C, \delta)$-anti-concentrated distribution if there is a univariate polynomial $p$ satisfying $p(0) = 1$ such that there is a degree $k$ sum-of-squares proof of the following two inequalities:

1. $\forall v, \langle Y, v \rangle^2 \leq \delta^2 \mathbb{E} \langle Y, v \rangle^2$ implies $(p(\langle Y, v \rangle) - 1)^2 \leq \delta^2$.

2. $\forall v, \|v\|_2^2 \leq 1$ implies $\mathbb{E} p^2(\langle Y, v \rangle) \leq C\delta$.

Intuitively, certified anti-concentration asks for a *certificate* of the anti-concentration property of $Y$ in the "sum-of-squares" proof system (see Section 3 for precise definitions). SoS is a proof system that

---

Please note that sections 3-6 are in the supplementary material.

reasons about polynomial inequalities. Since the "core indicator" $\mathbf{1}(|\langle x, v\rangle| \leq \delta)$ is not a polynomial, we phrase the condition in terms of an approximating polynomial $p$. We are now ready to state our main result.

**Theorem 1.5** (List-Decodable Regression). *For every $\alpha, \eta > 0$ and a $k$-certifiably $(C, \alpha^2\eta^2/10C)$-anti-concentrated distribution $D$ on $\mathbb{R}^d$, there exists an algorithm that takes input a sample generated according to $\mathrm{Lin}_D(\alpha, \ell^*)$ and outputs a list $L$ of size $O(1/\alpha)$ such that there is an $\ell \in L$ satisfying $\|\ell - \ell^*\|_2 < \eta$ with probability at least $0.99$ over the draw of the sample. The algorithm needs a sample of size $n = (kd)^{O(k)}$ and runs in time $n^{O(k)} = (kd)^{O(k^2)}$.*

*Remark* 1.6 (Tolerating Additive Noise). For additive noise (not necessarily independent across samples) of variance $\zeta^2$ in the inlier labels, our algorithm, in the same running time and sample complexity, outputs a list of size $O(1/\alpha)$ that contains an $\ell$ satisfying $\|\ell - \ell^*\|_2 \leq \frac{\zeta}{\alpha} + \eta$. Since we normalize $\ell^*$ to have unit norm, this guarantee is meaningful only when $\zeta \ll \alpha$.

*Remark* 1.7 (Exponential Dependence on $1/\alpha$). List-decodable regression algorithms immediately yield algorithms for mixed linear regression (MLR) without any assumptions on the components. The state-of-the-art algorithms for MLR with gaussian components [34, 41] has an exponential dependence on $k = 1/\alpha$ in the running time in the absence of strong pairwise separation or small condition number of the components. Liang and Liu [34] (see Page 10 of their paper) use the relationship to learning mixtures of $k$ gaussians (with an $\exp(k)$ lower bound [38]) to note that there may not exist any algorithms with polynomial dependence on $1/\alpha$ for MLR and thus, also for list-decodable regression.

**Certifiably anti-concentrated distributions**   In Section 5, we show certifiable anti-concentration of some well-studied families of distributions. This includes the standard gaussian distribution and more generally any anti-concentrated spherically symmetric distribution with strictly sub-exponential tails. We also show that simple operations such as scaling, applying well-conditioned linear transformations and sampling preserve certifiable anti-concentration. This yields:

**Corollary 1.8** (List-Decodable Regression for Gaussian Inliers). *For every $\alpha, \eta > 0$ there's an algorithm for list-decodable regression for the model $\mathrm{Lin}_D(\alpha, \ell^*)$ with $D = \mathcal{N}(0, \Sigma)$ with $\lambda_{\max}(\Sigma)/\lambda_{min}(\Sigma) = O(1)$ that needs $n = (d/\alpha\eta)^{O\left(\frac{1}{\alpha^4\eta^4}\right)}$ samples and runs in time $n^{O\left(\frac{1}{\alpha^4\eta^4}\right)} = (d/\alpha\eta)^{O\left(\frac{1}{\alpha^8\eta^8}\right)}$.*

We note that certifiably anti-concentrated distributions are more restrictive compared to the families of distributions for which the most general robust estimation algorithms work [31, 30, 28]. To a certain extent, this is inherent. The families of distributions considered in these prior works do not satisfy anti-concentration in general. And as we discuss in more detail in Section 2, anti-concentration is information-theoretically *necessary* (see Theorem 1.3) for list-decodable regression. This surprisingly rules out families of distributions that might appear natural and "easy", for example, the uniform distribution on $\{0, 1\}^n$.

We rescue this to an extent for the special case when $\ell^*$ in the model $\mathrm{Lin}(\alpha, \ell^*)$ is a "Boolean vector", i.e., has all coordinates of equal magnitude. Intuitively, this helps because while the the uniform distribution on $\{0, 1\}^n$ (and more generally, any discrete product distribution) is badly anti-concentrated in sparse directions, they are well anti-concentrated [20] in the directions that are far from any sparse vectors.

As before, for obtaining efficient algorithms, we need to work with a *certified* version (see Definition 4.5) of such a restricted anti-concentration condition. As a specific Corollary (see Theorem 4.6 for a more general statement), this allows us to show:

**Theorem 1.9** (List-Decodable Regression for Hypercube Inliers). *For every $\alpha, \eta > 0$ there's an $\eta$-approximate algorithm for list-decodable regression for the model $\mathrm{Lin}_D(\alpha, \ell^*)$ with $D$ is uniform on $\{0, 1\}^d$ that needs $n = (d/\alpha\eta)^{O(\frac{1}{\alpha^4\eta^4})}$ samples and runs in time $n^{O(\frac{1}{\alpha^4\eta^4})} = (d/\alpha\eta)^{O(\frac{1}{\alpha^8\eta^8})}$.*

In Section 4.1, we obtain similar results for general product distributions. It is an important open problem to prove certified anti-concentration for a broader family of distributions.

---

**Please note that sections** $3$-$6$ **are in the supplementary material.**

## 2 Overview of our Technique

In this section, we give a bird's eye view of our approach and illustrate the important ideas in our algorithm for list-decodable regression. Thus, given a sample $\mathcal{S} = \{(x_i, y_i)\}_{i=1}^n$ from $\mathrm{Lin}_D(\alpha, \ell^*)$, we must construct a constant-size list $L$ of linear functions containing an $\ell$ close to $\ell^*$.

Our algorithm is based on the sum-of-squares method. We build on the "identifiability to algorithms" paradigm developed in several prior works [5, 4, 36, 31, 24, 30, 28] with some important conceptual differences.

**An *inefficient* algorithm** Let's start by designing an inefficient algorithm for the problem. This may seem simple at the outset. But as we'll see, solving this relaxed problem will rely on some important conceptual ideas that will serve as a starting point for our efficient algorithm.

Without computational constraints, it is natural to just return the list $L$ of all linear functions $\ell$ that correctly labels all examples in some $S \subseteq \mathcal{S}$ of size $\alpha n$. We call such an $S$, a large, *soluble* set. True inliers $\mathcal{I}$ satisfy our search criteria so $\ell^* \in L$. However, it's not hard to show (Proposition B.1 ) that one can choose outliers so that the list so generated has size $\exp(d)$ (far from a fixed constant!).

A potential fix is to search instead for a *coarse soluble partition* of $\mathcal{S}$, if it exists, into disjoint $S_1, S_2, \ldots, S_k$ and linear functions $\ell_1, \ell_2, \ldots, \ell_k$ so that every $|S_i| \geq \alpha n$ and $\ell_i$ correctly computes the labels in $S_i$. In this setting, our list is small ($k \leq 1/\alpha$). But it is easy to construct samples $\mathcal{S}$ for which this fails because there are coarse soluble partitions of $\mathcal{S}$ where every $\ell_i$ is far from $\ell^*$.

**Anti-Concentration** It turns out that any (even inefficient) algorithm for list-decodable regression provably (see Theorem 6.1) *requires* that the distribution of inliers[2] be sufficiently *anti-concentrated*:

**Definition 2.1** (Anti-Concentration). A $\mathbb{R}^d$-valued random variable $Y$ with mean 0 is $\delta$-anti-concentrated[3] if for all non-zero $v$, $\Pr[\langle Y, v \rangle = 0] < \delta$. A set $T \subseteq \mathbb{R}^d$ is $\delta$-anti-concentrated if the uniform distribution on $T$ is $\delta$-anti-concentrated.

As we discuss next, anti-concentration is also *sufficient* for list-decodable regression. Intuitively, this is because anti-concentration of the inliers prevents the existence of a soluble set that intersects significantly with $\mathcal{I}$ and yet can be labeled correctly by $\ell \neq \ell^*$. This is simple to prove in the special case when $\mathcal{S}$ admits a coarse soluble partition.

**Proposition 2.2.** *Suppose $\mathcal{I}$ is $\alpha$-anti-concentrated. Suppose there exists a partition $S_1, S_2, \ldots, S_k \subseteq \mathcal{S}$ such that each $|S_i| \geq \alpha n$ and there exist $\ell_1, \ell_2, \ldots, \ell_k$ such that $y_j = \langle \ell_i, x_j \rangle$ for every $j \in S_i$. Then, there is an $i$ such that $\ell_i = \ell^*$.*

*Proof.* Since $k \leq 1/\alpha$, there is a $j$ such that $|\mathcal{I} \cap S_j| \geq \alpha|\mathcal{I}|$. Then, $\langle x_i, \ell_j \rangle = \langle x_i, \ell^* \rangle$ for every $i \in \mathcal{I} \cap S_j$. Thus, $\Pr_{i \sim \mathcal{I}}[\langle x_i, \ell_j - \ell^* \rangle = 0] \geq \alpha$. This contradicts anti-concentration of $\mathcal{I}$ unless $\ell_j - \ell^* = 0$. $\square$

The above proposition allows us to use *any* soluble partition as a *certificate* of correctness for the associated list $L$. Two aspects of this certificate were crucial in the above argument: 1) *largeness*: each $S_i$ is of size $\alpha n$ - so the generated list is small, and, 2) *uniformity*: every sample is used in exactly one of the sets so $\mathcal{I}$ must intersect one of the $S_i$s in at least $\alpha$-fraction of the points.

**Identifiability via anti-concentration** For arbitrary $\mathcal{S}$, a coarse soluble partition might not exist. So we will generalize coarse soluble partitions to obtain certificates that exist for every sample $\mathcal{S}$ and guarantee largeness and a relaxation of uniformity (formalized below). For this purpose, it is convenient to view such certificates as distributions $\mu$ on $\geq \alpha n$ size soluble subsets of $\mathcal{S}$ so any collection $\mathcal{C} \subseteq 2^\mathcal{S}$ of $\alpha n$ size sets corresponds to the uniform distribution $\mu$ on $\mathcal{C}$.

To precisely define uniformity, let $W_i(\mu) = \mathbb{E}_{S \sim \mu}[\mathbf{1}(i \in S)]$ be the "frequency of i", that is, probability that the $i$th sample is chosen to be in a set drawn according to $\mu$. Then, the uniform distribution $\mu$ on any coarse soluble $k$-partition satisfies $W_i = \frac{1}{k}$ for every $i$. That is, all samples

---

**Please note that sections** 3-6 **are in the supplementary material.**

[2]As in the standard robust estimation setting, the outliers are arbitrary and potentially adversarially chosen.

[3]Definition 1.4 differs slightly to handle list-decodable regression with additive noise in the inliers.

$i \in \mathcal{S}$ are *uniformly* used in such a $\mu$. To generalize this idea, we define $\sum_i W_i(\mu)^2$ as the *distance to uniformity* of $\mu$. Up to a shift, this is simply the variance in the frequencies of the points in $\mathcal{S}$ used in draws from $\mu$. Our generalization of a coarse soluble partition of $\mathcal{S}$ is any $\mu$ that minimizes $\sum_i W_i(\mu)^2$, the distance to uniformity, and is thus *maximally uniform* among all distributions supported on large soluble sets. Such a $\mu$ can be found by convex programming.

The following claim generalizes Proposition 2.2 to derive the same conclusion starting from any maximally uniform distribution supported on large soluble sets.

**Proposition 2.3.** *For a maximally uniform $\mu$ on $\alpha n$ size soluble subsets of $\mathcal{S}$, $\sum_{i \in \mathcal{I}} \mathbb{E}_{S \sim \mu}[\mathbf{1}(i \in S)] \geq \alpha|\mathcal{I}|$.*

The proof proceeds by contradiction (see Lemma 4.3). We show that if $\sum_{i \in \mathcal{I}} W_i(\mu) \leq \alpha|\mathcal{I}|$, then we can strictly reduce the distance to uniformity by taking a mixture of $\mu$ with the distribution that places all its probability mass on $\mathcal{I}$. This allow us to obtain an (inefficient) algorithm for list-decodable regression establishing identifiability.

**Proposition 2.4** (Identifiability for List-Decodable Regression). *Let $\mathcal{S}$ be sample from $\mathrm{Lin}(\alpha, \ell^*)$ such that $\mathcal{I}$ is $\delta$-anti-concentrated for $\delta < \alpha$. Then, there's an (inefficient) algorithm that finds a list $L$ of size $\frac{20}{\alpha-\delta}$ such that $\ell^* \in L$ with probability at least $0.99$.*

*Proof.* Let $\mu$ be *any* maximally uniform distribution over $\alpha n$ size soluble subsets of $\mathcal{S}$. For $k = \frac{20}{\alpha-\delta}$, let $S_1, S_2, \ldots, S_k$ be independent samples from $\mu$. Output the list $L$ of $k$ linear functions that correctly compute the labels in each $S_i$.

To see why $\ell^* \in L$, observe that $\mathbb{E}|S_j \cap \mathcal{I}| = \sum_{i \in \mathcal{I}} \mathbb{E}\mathbf{1}(i \in S_j) \geq \alpha|\mathcal{I}|$. By averaging, $\Pr[|S_j \cap \mathcal{I}| \geq \frac{\alpha+\delta}{2}|\mathcal{I}|] \geq \frac{\alpha-\delta}{2}$. Thus, there's a $j \leq k$ so that $|S_j \cap \mathcal{I}| \geq \frac{\alpha+\delta}{2}|\mathcal{I}|$ with probability at least $1 - (1 - \frac{\alpha-\delta}{2})^{\frac{20}{\alpha-\delta}} \geq 0.99$. We can now repeat the argument in the proof of Proposition 2.2 to conclude that any linear function that correctly labels $S_j$ must equal $\ell^*$. $\qquad\square$

**An efficient algorithm**  Our identifiability proof suggests the following simple algorithm: 1) find *any* maximally uniform distribution $\mu$ on soluble subsets of size $\alpha n$ of $\mathcal{S}$, 2) take $O(1/\alpha)$ samples $S_i$ from $\mu$ and 3) return the list of linear functions that correctly label the equations in $S_i$s. This is inefficient because searching over distributions is NP-hard in general.

To make this into an efficient algorithm, we start by observing that soluble subsets $S \subseteq \mathcal{S}$ of size $\alpha n$ can be described by the following set of quadratic equations where $w$ stands for the indicator of $S$ and $\ell$, the linear function that correctly labels the examples in $S$.

$$\mathcal{A}_{w,\ell}: \begin{cases} \sum_{i=1}^n w_i = \alpha n \\ \forall i \in [n]. \quad\quad\quad\quad\quad w_i^2 = w_i \\ \forall i \in [n]. \quad w_i \cdot (y_i - \langle x_i, \ell \rangle) = 0 \\ \|\ell\|^2 \leq 1 \end{cases} \tag{2.1}$$

Our efficient algorithm searches for a maximally uniform *pseudo-distribution* on $w$ satisfying (2.1). Degree $k$ pseudo-distributions (see Section 3 for precise definitions) are generalization of distributions that nevertheless "behave" just as distributions whenever we take (pseudo)-expectations (denoted by $\tilde{\mathbb{E}}$) of a class of degree $k$ polynomials. And unlike distributions, degree $k$ pseudo-distributions satisfying[4] polynomial constraints (such as (2.1)) can be computed in time $n^{O(k)}$.

For the sake of intuition, it might be helpful to (falsely) think of pseudo-distributions $\tilde{\mu}$ as simply distributions where we only get access to moments of degree $\leq k$. Thus, we are allowed to compute expectations of all degree $\leq k$ polynomials with respect to $\tilde{\mu}$. Since $W_i(\tilde{\mu}) = \tilde{\mathbb{E}}_{\tilde{\mu}} w_i$ are just first moments of $\tilde{\mu}$, our notion of maximally uniform distributions extends naturally to pseudo-distributions. This allows us to prove an analog of Proposition 2.3 for pseudo-distributions and gives us an efficient replacement for Step 1.

---

**Please note that sections** 3-6 **are in the supplementary material.**

[4]See Fact 3.3 for a precise statement.

**Proposition 2.5.** *For any maximally uniform $\tilde{\mu}$ of degree $\geq$ 2, $\sum_{i \in \mathcal{I}} \tilde{\mathbb{E}}_{\tilde{\mu}}[w_i] \geq \alpha|\mathcal{I}| = \alpha \sum_{i \in [n]} \tilde{\mathbb{E}}_{\tilde{\mu}}[w_i]$.*

For Step 2, however, we hit a wall: it's not possible to obtain independent samples from $\tilde{\mu}$ given only low-degree moments.

**Rounding by Votes**    To circumvent this hurdle, our algorithm departs from rounding strategies for pseudo-distributions used in prior works and instead "rounds" *each* sample to a candidate linear function. While a priori, this method produces $n$ different candidates instead of one, we will be able to extract a list of $O(\frac{1}{\alpha})$ size that contains the true vector from them. This step will crucially rely on anti-concentration properties of $\mathcal{I}$.

Consider the vector $v_i = \frac{\tilde{\mathbb{E}}_{\tilde{\mu}}[w_i \ell]}{\tilde{\mathbb{E}}_{\tilde{\mu}}[w_i]}$ whenever $\tilde{\mathbb{E}}_{\tilde{\mu}}[w_i] \neq 0$ (set $v_i$ to zero, otherwise). This is simply the (scaled) average, according to $\tilde{\mu}$, of all the linear functions $\ell$ that are used to label the sets $S$ of size $\alpha n$ in the support of $\tilde{\mu}$ whenever $i \in S$. Further, $v_i$ depends only on the first two moments of $\tilde{\mu}$.

We think of $v_i$s as "votes"cast by the $i$th sample for the unknown linear function. Let us focus our attention on the votes $v_i$ of $i \in \mathcal{I}$ - the inliers. We will show that according to the distribution proportional to $\tilde{\mathbb{E}}[w]$, the average $\ell_2$ distance of $v_i$ from $\ell^*$ is at max $\eta$:

$$\frac{1}{\sum_{i \in \mathcal{I}} \tilde{\mathbb{E}}[w_i]} \sum_{i \in \mathcal{I}} \tilde{\mathbb{E}}[w_i] \|v_i - \ell^*\|_2 < \eta. \tag{$\star$}$$

Before diving into ($\star$), let's see how it gives us our efficient list-decodable regression algorithm:

1. Find a pseudo-distribution $\tilde{\mu}$ satisfying (2.1) that minimizes distance to uniformity $\sum_i \tilde{\mathbb{E}}_{\tilde{\mu}}[w_i]^2$.

2. For $O(\frac{1}{\alpha})$ times, independently choose a random index $i \in [n]$ with probability proportional to $\tilde{\mathbb{E}}_{\tilde{\mu}}[w_i]$ and return the list of corresponding $v_i$s.

Step 1 above is a convex program - it minimizes a norm subject on the convex set of pseudo-distributions - and can be solved in polynomial time. Let's analyze step 2 to see why the algorithm works. Using ($\star$) and Markov's inequality, conditioned on $i \in \mathcal{I}$, $\|v_i - \ell^*\|_2 \leq 2\eta$ with probability $\geq 1/2$. By Proposition 2.5, $\frac{\sum_{i \in \mathcal{I}} \tilde{\mathbb{E}}[w_i]}{\sum_{i \in [n]} \tilde{\mathbb{E}}[w_i]} \geq \alpha$ so $i \in \mathcal{I}$ with probability at least $\alpha$. Thus in each iteration of step 2, with probability at least $\alpha/2$, we choose an $i$ such that $v_i$ is $2\eta$-close to $\ell^*$. Repeating $O(1/\alpha)$ times gives us the 0.99 chance of success.

($\star$) **via anti-concentration**    As in the information-theoretic argument, ($\star$) relies on the anti-concentration of $\mathcal{I}$. Let's do a quick proof for the case when $\tilde{\mu}$ is an actual distribution $\mu$.

*Proof of ($\star$) for actual distributions $\mu$.*  Observe that $\mu$ is a distribution over $(w, \ell)$ satisfying (2.1). Recall that $w$ indicates a subset $S \subseteq \mathcal{S}$ of size $\alpha n$ and $w_i = 1$ iff $i \in S$. And $\ell \in \mathbb{R}^d$ satisfies all the equations in $S$.

By Cauchy-Schwarz, $\sum_i \|\mathbb{E}_\mu[w_i \ell] - \mathbb{E}_\mu[w_i]\ell^*\| \leq \mathbb{E}_\mu[\sum_{i \in \mathcal{I}} w_i \|\ell - \ell^*\|]$. Next, as in Proposition 2.2, since $\mathcal{I}$ is $\eta$-anti-concentrated, and for all $S$ such that $|\mathcal{I} \cap S| \geq \eta|\mathcal{I}|$, $\ell - \ell^* = 0$. Thus, any such $S$ in the support of $\mu$ contributes 0 to the expectation above. We will now show that the contribution from the remaining terms is upper bounded by $\eta$. Observe that since $\|\ell - \ell^*\| \leq 2$, $\mathbb{E}_\mu[\sum_{i \in \mathcal{I}} w_i \|\ell - \ell^*\|] = \mathbb{E}_\mu[\mathbf{1}\left(|S \cap \mathcal{I}| < \eta|\mathcal{I}|\right) w_i \|\ell - \ell^*\|] = \mathbb{E}_\mu[\sum_{i \in S \cap \mathcal{I}} \|\ell - \ell^*\|] \leq 2\eta|\mathcal{I}|.$  $\square$

**SoSizing Anti-Concentration**    The key to proving ($\star$) for pseudo-distributions is a *sum-of-squares* (SoS) proof of anti-concentration inequality: $\Pr_{x \sim \mathcal{I}}[\langle x, v \rangle = 0] \leq \eta$ in variable $v$. SoS is a restricted system for proving polynomial inequalities subject to polynomial inequality constraints. Thus, to even ask for a SoS proof we must phrase anti-concentration as a polynomial inequality.

---

**Please note that sections 3-6 are in the supplementary material.**

To do this, let $p(z)$ be a low-degree polynomial approximator for the function $\mathbf{1}\,(z = 0)$. Then, we can hope to "replace" the use of the inequality $\Pr_{x \sim \mathcal{I}}[\langle x, v \rangle = 0] \leq \eta \equiv \mathbb{E}_{x \sim \mathcal{I}}[\mathbf{1}(\langle x, v \rangle = 0)] \leq \eta$ in the argument above by $\mathbb{E}_{x \sim \mathcal{I}}[p(\langle x, v \rangle)^2] \leq \eta$. Since polynomials grow unboundedly for large enough inputs, it is *necessary* for the uniform distribution on $\mathcal{I}$ to have sufficiently light-tails to ensure that $\mathbb{E}_{x \sim \mathcal{I}} p(\langle x, v \rangle)^2$ is small. In Lemma A.1, we show that anti-concentration and strictly sub-exponential tails are *sufficient* to construct such a polynomial.

We can finally ask for a SoS proof for $\mathbb{E}_{x \sim \mathcal{I}} p(\langle x, v \rangle) \leq \eta$ in variable $v$. We prove such *certified* anti-concentration inequalities for broad families of inlier distributions in Section 5.

## 3  Acknowledgements

The authors would like to thank the following sources of support.

Sushrut Karmalkar was supported by NSF Award CNS-1414023. Adam Klivans was supported by NSF Award CCF-1717896. Pravesh Kothari was supported by Schmidt Foundation Fellowship and Avi Wigderson's NSF Award CCF-1412958.

## Footnotes

[1]There's a long line of work on robust regression algorithms (see for e.g. [7, 27]) that can tolerate corruptions only in the *labels*. We are interested in algorithms robust against corruptions in both examples and labels.

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
