[Supplementary Material · supplementary.pdf]

# List-decodable Linear Regression

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

# 3 Preliminaries

313 In this section, we define pseudo-distributions and sum-of-squares proofs. See the lecture notes [6]
314 for more details and the appendix in [44] for proofs of the propositions appearing here.

315 Let $x = (x_1, x_2, \ldots, x_n)$ be a tuple of $n$ indeterminates and let $\mathbb{R}[x]$ be the set of polynomials
316 with real coefficients and indeterminates $x_1, \ldots, x_n$. We say that a polynomial $p \in \mathbb{R}[x]$ is a
317 *sum-of-squares (sos)* if there are polynomials $q_1, \ldots, q_r$ such that $p = q_1^2 + \cdots + q_r^2$.

## 3.1 Pseudo-distributions

319 Pseudo-distributions are generalizations of probability distributions. We can represent a discrete (i.e.,
320 finitely supported) probability distribution over $\mathbb{R}^n$ by its probability mass function $D : \mathbb{R}^n \to \mathbb{R}$
321 such that $D \geq 0$ and $\sum_{x \in \operatorname{supp}(D)} D(x) = 1$. Similarly, we can describe a pseudo-distribution by its
322 mass function. Here, we relax the constraint $D \geq 0$ and only require that $D$ passes certain low-degree
323 non-negativity tests.

324 Concretely, a *level-$\ell$ pseudo-distribution* is a finitely-supported function $D : \mathbb{R}^n \to \mathbb{R}$ such that
325 $\sum_x D(x) = 1$ and $\sum_x D(x) f(x)^2 \geq 0$ for every polynomial $f$ of degree at most $\ell/2$. (Here, the
326 summations are over the support of $D$.) A straightforward polynomial-interpolation argument shows
327 that every level-$\infty$-pseudo distribution satisfies $D \geq 0$ and is thus an actual probability distribution.
328 We define the *pseudo-expectation* of a function $f$ on $\mathbb{R}^d$ with respect to a pseudo-distribution $D$,
329 denoted $\tilde{\mathbb{E}}_{D(x)} f(x)$, as

$$\tilde{\mathbb{E}}_{D(x)} f(x) = \sum_x D(x) f(x) . \tag{3.1}$$

330 The degree-$\ell$ moment tensor of a pseudo-distribution $D$ is the tensor $\mathbb{E}_{D(x)}(1, x_1, x_2, \ldots, x_n)^{\otimes \ell}$. In
331 particular, the moment tensor has an entry corresponding to the pseudo-expectation of all monomials
332 of degree at most $\ell$ in $x$. The set of all degree-$\ell$ moment tensors of probability distribution is a
333 convex set. Similarly, the set of all degree-$\ell$ moment tensors of degree $d$ pseudo-distributions is also
334 convex. Key to the algorithmic utility of pseudo-distributions is the fact that while there can be no
335 efficient separation oracle for the convex set of all degree-$\ell$ moment tensors of an actual probability
336 distribution, there's a separation oracle running in time $n^{O(\ell)}$ for the convex set of the degree-$\ell$
337 moment tensors of all level-$\ell$ pseudodistributions.

338 **Fact 3.1** ([52, 48, 47, 40]). *For any $n, \ell \in \mathbb{N}$, the following set has a $n^{O(\ell)}$-time weak separation*
339 *oracle (in the sense of [28]):*

$$\left\{ \tilde{\mathbb{E}}_{D(x)}(1, x_1, x_2, \ldots, x_n)^{\otimes d} \mid \text{degree-d pseudo-distribution } D \text{ over } \mathbb{R}^n \right\} . \tag{3.2}$$

340 This fact, together with the equivalence of weak separation and optimization [28] allows us to
341 efficiently optimize over pseudo-distributions (approximately)—this algorithm is referred to as the
342 sum-of-squares algorithm.

343 The *level-$\ell$ sum-of-squares algorithm* optimizes over the space of all level-$\ell$ pseudo-distributions that
344 satisfy a given set of polynomial constraints—we formally define this next.

345 **Definition 3.2** (Constrained pseudo-distributions). Let $D$ be a level-$\ell$ pseudo-distribution over $\mathbb{R}^n$.
346 Let $\mathcal{A} = \{ f_1 \geq 0, f_2 \geq 0, \ldots, f_m \geq 0 \}$ be a system of $m$ polynomial inequality constraints. We say
347 that $D$ *satisfies the system of constraints $\mathcal{A}$ at degree $r$*, denoted $D \models_{\overline{r}} \mathcal{A}$, if for every $S \subseteq [m]$ and
348 every sum-of-squares polynomial $h$ with $\deg h + \sum_{i \in S} \max\{\deg f_i, r\}$,

$$\tilde{\mathbb{E}}_D h \cdot \prod_{i \in S} f_i \geq 0 .$$

We write $D \models \mathcal{A}$ (without specifying the degree) if $D \models_{\overline{0}} \mathcal{A}$ holds. Furthermore, we say that $D \models_{\overline{r}} \mathcal{A}$ holds *approximately* if the above inequalities are satisfied up to an error of $2^{-n^\ell} \cdot \|h\| \cdot \prod_{i \in S} \|f_i\|$, where $\|\cdot\|$ denotes the Euclidean norm[5] of the coefficients of a polynomial in the monomial basis.

We remark that if $D$ is an actual (discrete) probability distribution, then we have $D \models \mathcal{A}$ if and only if $D$ is supported on solutions to the constraints $\mathcal{A}$.

We say that a system $\mathcal{A}$ of polynomial constraints is *explicitly bounded* if it contains a constraint of the form $\{\|x\|^2 \leq M\}$. The following fact is a consequence of Fact 3.1 and [28],

**Fact 3.3** (Efficient Optimization over Pseudo-distributions). *There exists an $(n + m)^{O(\ell)}$-time algorithm that, given any explicitly bounded and satisfiable system[6] $\mathcal{A}$ of $m$ polynomial constraints in $n$ variables, outputs a level-$\ell$ pseudo-distribution that satisfies $\mathcal{A}$ approximately.*

### 3.2 Sum-of-squares proofs

Let $f_1, f_2, \ldots, f_r$ and $g$ be multivariate polynomials in $x$. A *sum-of-squares proof* that the constraints $\{f_1 \geq 0, \ldots, f_m \geq 0\}$ imply the constraint $\{g \geq 0\}$ consists of polynomials $(p_S)_{S \subseteq [m]}$ such that

$$g = \sum_{S \subseteq [m]} p_S \cdot \Pi_{i \in S} f_i . \tag{3.3}$$

We say that this proof has *degree* $\ell$ if for every set $S \subseteq [m]$, the polynomial $p_S \Pi_{i \in S} f_i$ has degree at most $\ell$. If there is a degree $\ell$ SoS proof that $\{f_i \geq 0 \mid i \leq r\}$ implies $\{g \geq 0\}$, we write:

$$\{f_i \geq 0 \mid i \leq r\} \vdash_{\ell} \{g \geq 0\} . \tag{3.4}$$

Sum-of-squares proofs satisfy the following inference rules. For all polynomials $f, g \colon \mathbb{R}^n \to \mathbb{R}$ and for all functions $F \colon \mathbb{R}^n \to \mathbb{R}^m$, $G \colon \mathbb{R}^n \to \mathbb{R}^k$, $H \colon \mathbb{R}^p \to \mathbb{R}^n$ such that each of the coordinates of the outputs are polynomials of the inputs, we have:

$$\frac{\mathcal{A} \vdash_{\ell} \{f \geq 0, g \geq 0\}}{\mathcal{A} \vdash_{\ell} \{f + g \geq 0\}}, \frac{\mathcal{A} \vdash_{\ell} \{f \geq 0\}, \mathcal{A} \vdash_{\ell'} \{g \geq 0\}}{\mathcal{A} \vdash_{\ell+\ell'} \{f \cdot g \geq 0\}} \qquad \text{(addition and multiplication)}$$

$$\frac{\mathcal{A} \vdash_{\ell} \mathcal{B}, \mathcal{B} \vdash_{\ell'} C}{\mathcal{A} \vdash_{\ell \cdot \ell'} C} \qquad \text{(transitivity)}$$

$$\frac{\{F \geq 0\} \vdash_{\ell} \{G \geq 0\}}{\{F(H) \geq 0\} \vdash_{\ell \cdot \deg(H)} \{G(H) \geq 0\}} . \qquad \text{(substitution)}$$

Low-degree sum-of-squares proofs are sound and complete if we take low-level pseudo-distributions as models.

Concretely, sum-of-squares proofs allow us to deduce properties of pseudo-distributions that satisfy some constraints.

**Fact 3.4** (Soundness). *If $D \models_{\overline{r}} \mathcal{A}$ for a level-$\ell$ pseudo-distribution $D$ and there exists a sum-of-squares proof $\mathcal{A} \vdash_{r'} \mathcal{B}$, then $D \models_{\overline{r \cdot r' + r'}} \mathcal{B}$.*

If the pseudo-distribution $D$ satisfies $\mathcal{A}$ only approximately, soundness continues to hold if we require an upper bound on the bit-complexity of the sum-of-squares $\mathcal{A} \vdash_{r'} B$ (number of bits required to write down the proof).

In our applications, the bit complexity of all sum of squares proofs will be $n^{O(\ell)}$ (assuming that all numbers in the input have bit complexity $n^{O(1)}$). This bound suffices in order to argue about pseudo-distributions that satisfy polynomial constraints approximately.

The following fact shows that every property of low-level pseudo-distributions can be derived by low-degree sum-of-squares proofs.

**Fact 3.5** (Completeness)**.** *Suppose $d \geq r' \geq r$ and $\mathcal{A}$ is a collection of polynomial constraints with degree at most $r$, and $\mathcal{A} \vdash \{\sum_{i=1}^{n} x_i^2 \leq B\}$ for some finite $B$.*

*Let $\{g \geq 0\}$ be a polynomial constraint. If every degree-$d$ pseudo-distribution that satisfies $D \mathrel{\models_{\overline{r}}} \mathcal{A}$ also satisfies $D \mathrel{\models_{\overline{r'}}} \{g \geq 0\}$, then for every $\epsilon > 0$, there is a sum-of-squares proof $\mathcal{A} \mathrel{\mid_{\overline{d}}} \{g \geq -\epsilon\}$.*

We will use the following Cauchy-Schwarz inequality for pseudo-distributions:

**Fact 3.6** (Cauchy-Schwarz for Pseudo-distributions)**.** *Let $f, g$ be polynomials of degree at most $d$ in indeterminate $x \in \mathbb{R}^d$. Then, for any degree $d$ pseudo-distribution $\tilde{\mu}$, $\tilde{\mathbb{E}}_{\tilde{\mu}}[fg] \leq \sqrt{\tilde{\mathbb{E}}_{\tilde{\mu}}[f^2]}\sqrt{\tilde{\mathbb{E}}_{\tilde{\mu}}[g^2]}$.*

The following fact is a simple corollary of the fundamental theorem of algebra:

**Fact 3.7.** *For any univariate degree $d$ polynomial $p(x) \geq 0$ for all $x \in \mathbb{R}$, $\mathrel{\mid_{\overline{d}}^{x}} \{p(x) \geq 0\}$.*

This can be extended to univariate polynomial inequalities over intervals of $\mathbb{R}$.

**Fact 3.8** (Fekete and Markov-Lukács, see [41])**.** *For any univariate degree $d$ polynomial $p(x) \geq 0$ for $x \in [a,b]$, $\{x \geq a, x \leq b\} \mathrel{\mid_{\overline{d}}^{x}} \{p(x) \geq 0\}$.*

# 4 Algorithm for List-Decodable Robust Regression

In this section, we describe and analyze our algorithm for list-decodable regression and prove our first main result restated here.

**Theorem 1.5** (List-Decodable Regression)**.** *For every $\alpha, \eta > 0$ and a $k$-certifiably $(C, \alpha^2\eta^2/10C)$-anti-concentrated distribution $D$ on $\mathbb{R}^d$, there exists an algorithm that takes input a sample generated according to $\mathrm{Lin}_D(\alpha, \ell^*)$ and outputs a list $L$ of size $O(1/\alpha)$ such that there is an $\ell \in L$ satisfying $\|\ell - \ell^*\|_2 < \eta$ with probability at least $0.99$ over the draw of the sample. The algorithm needs a sample of size $n = (kd)^{O(k)}$ and runs in time $n^{O(k)} = (kd)^{O(k^2)}$.*

We will analyze Algorithm 1 to prove Theorem 1.5.

$$
\mathcal{A}_{w,\ell} \colon \begin{cases}
\hspace{3.5cm} \sum_{i=1}^{n} w_i = \alpha n \\
\forall i \in [n]. \hspace{2.3cm} w_i^2 = w_i \\
\forall i \in [n]. \quad w_i \cdot (y_i - \langle x_i, \ell \rangle) = 0 \\
\hspace{3cm} \sum_{i \leq d} \ell_i^2 \leq 1
\end{cases} \tag{4.1}
$$

---

**Algorithm 1** (List-Decodable Regression)**.**

**Given:** Sample $\mathcal{S}$ of size $n$ drawn according to $\mathrm{Lin}(\alpha, n, \ell^*)$ with inliers $\mathcal{I}$, $\eta > 0$.

**Output:** A list $L \subseteq \mathbb{R}^d$ of size $O(1/\alpha)$ such that there exists a $\ell \in L$ satisfying $\|\ell - \ell^*\|_2 < \eta$.

**Operation:**

1. Find a degree $O(1/\alpha^4\eta^4)$ pseudo-distribution $\tilde{\mu}$ satisfying $\mathcal{A}_{w,\ell}$ that minimizes $\|\tilde{\mathbb{E}}[w]\|_2$.
2. For each $i \in [n]$ such that $\tilde{\mathbb{E}}_{\tilde{\mu}}[w_i] > 0$, let $v_i = \frac{\tilde{\mathbb{E}}_{\tilde{\mu}}[w_i \ell]}{\tilde{\mathbb{E}}_{\tilde{\mu}}[w_i]}$. Otherwise, set $v_i = 0$.
3. Take $J$ be a random multiset formed by union of $O(1/\alpha)$ independent draws of $i \in [n]$ with probability $\frac{\tilde{\mathbb{E}}[w_i]}{\alpha n}$.
4. Output $L = \{v_i \mid i \in J\}$ where $J \subseteq [n]$.

---

Our analysis follows the discussion in the overview. We start by formally proving ($\star$).

**Lemma 4.1.** *For any $t \geq k$ and any $\mathcal{S}$ so that $\mathcal{I} \subseteq \mathcal{S}$ is $k$-certifiably $(C, \alpha^2\eta^2/4C)$-anti-concentrated,*

$$\mathcal{A}_{w,\ell} \left|\frac{w,\ell}{t}\right. \left\{ \frac{1}{|\mathcal{I}|} \sum_{i \in \mathcal{I}}^{n} w_i \|\ell - \ell^*\|_2^2 \leq \frac{\alpha^2\eta^2}{4} \right\}$$

*Proof.* We start by observing:

$$\mathcal{A}_{w,\ell} \left|\frac{\ell}{2}\right. \|\ell - \ell^*\|_2^2 \leq 2 \,.$$

Since $\mathcal{I}$ is $(C, \alpha\eta/2C)$-anti-concentrated, there exists a univariate polynomial $p$ such that $\forall i$:

$$\{w_i\langle x, \ell - \ell^*\rangle = 0\} \left|\frac{k}{\ell}\right. \{p(w_i\langle x_i, \ell - \ell^*\rangle) = 1\} \tag{4.2}$$

and

$$\{\|\ell\|^2 \leq 1\} \left|\frac{k}{\ell}\right. \left\{ \frac{1}{|\mathcal{I}|} \sum_{i \in \mathcal{I}} p(\langle x_i, \ell - \ell^*\rangle)^2 \leq \frac{\alpha^2\eta^2}{4} \right\} \tag{4.3}$$

Using (4.2), we have:

$$\mathcal{A}_{w,\ell} \left|\frac{w,\ell}{t+2}\right. \left\{ 1 - p^2(w_i\langle x_i, \ell - \ell^*\rangle) = 0 \right\} \left|\frac{w,\ell}{t+2}\right. \left\{ 1 - w_i p^2(\langle x_i, \ell - \ell^*\rangle) = 0 \right\}$$

Using (4.3) and $\mathcal{A}_{w,\ell} \left|\frac{2}{w}\right. \{w_i^2 = w_i\}$, we thus have:

$$\mathcal{A}_{w,\ell} \left|\frac{w,\ell}{t+2}\right. \left\{ \frac{1}{|\mathcal{I}|} \sum_{i \in \mathcal{I}} w_i \|\ell - \ell^*\|_2^2 = \frac{1}{|\mathcal{I}|} \sum_{i \in \mathcal{I}} w_i \|\ell - \ell^*\|_2^2 w_i p^2(\langle x_i, \ell - \ell^*\rangle) = \frac{1}{|\mathcal{I}|} \sum_{i \in \mathcal{I}} w_i \|\ell - \ell^*\|_2^2 p^2(\langle x_i, \ell - \ell^*\rangle) \right.$$

$$\leq \frac{1}{|\mathcal{I}|} \sum_{i \in \mathcal{I}} \|\ell - \ell^*\|_2^2 p^2(\langle x_i, \ell - \ell^*\rangle) \leq \frac{\alpha^2\eta^2}{4} \,. \bigg\}$$

$\square$

As a consequence of this lemma, we can show that a constant fraction of the $v_i$ for $i \in \mathcal{I}$ constructed in the algorithm are close to $\ell^*$.

**Lemma 4.2.** *For any $\tilde{\mu}$ of degree $k$ satisfying $\mathcal{A}_{w,\ell}$, $\frac{1}{|\mathcal{I}|} \sum_{i \in \mathcal{I}} \tilde{\mathbb{E}}[w_i] \cdot \|v_i - \ell^*\|_2 \leq \frac{\alpha}{2}\eta$.*

*Proof.* By Lemma 4.1, we have: $\mathcal{A}_{w,\ell} \left|\frac{w,\ell}{k}\right. \left\{ \frac{1}{|\mathcal{I}|} \sum_{i \in \mathcal{I}}^{n} w_i \|\ell - \ell^*\|_2^2 \leq \frac{\alpha^2\eta^2}{4} \right\}$.

We also have: $\mathcal{A}_{w,\ell} \left|\frac{w,\ell}{2}\right. \{w_i^2 - w_i = 0\}$ for any $i$. This yields:

$$\mathcal{A}_{w,\ell} \left|\frac{w,\ell}{k}\right. \left\{ \frac{1}{|\mathcal{I}|} \sum_{i \in \mathcal{I}}^{n} \|w_i\ell - w_i\ell^*\|_2^2 \leq \frac{\alpha^2\eta^2}{4} \right\}$$

Since $\tilde{\mu}$ satisfies $\mathcal{A}_{w,\ell}$, taking pseudo-expectations yields: $\frac{1}{\mathcal{I}} \sum_{i \in \mathcal{I}} \tilde{\mathbb{E}} \|w_i\ell - w_i\ell^*\|_2^2 \leq \frac{\alpha^2\eta^2}{4}$.

By Cauchy-Schwarz for pseudo-distributions (Fact 3.6), we have:

$$\left( \frac{1}{\mathcal{I}} \sum_{i \in \mathcal{I}} \| \tilde{\mathbb{E}}[w_i\ell] - \tilde{\mathbb{E}}[w_i]\ell^* \|_2 \right)^2 \leq \frac{1}{\mathcal{I}} \sum_{i \in \mathcal{I}} \| \tilde{\mathbb{E}}[w_i\ell] - \tilde{\mathbb{E}}[w_i]\ell^* \|_2^2 \leq \frac{\alpha^2\eta^2}{4} \,.$$

Using $v_i = \frac{\tilde{\mathbb{E}}[w_i\ell]}{\tilde{\mathbb{E}}[w_i]}$ if $\tilde{\mathbb{E}}[w_i] > 0$ and 0 otherwise, we have: $\frac{1}{\mathcal{I}} \sum_{i \in \mathcal{I}, \tilde{\mathbb{E}}[w_i] > 0} \tilde{\mathbb{E}}[w_i] \cdot \|v_i - \ell^*\|_2 \leq \frac{\alpha}{2}\eta$.

$\square$

422 Next, we formally prove that maximally uniform pseudo-distributions satisfy Proposition 2.5.

423 **Lemma 4.3.** *For any $\tilde{\mu}$ of degree $\geq 4$ satisfying $\mathcal{A}_{w,\ell}$ that minimizes $\|\tilde{\mathbb{E}}[w]\|_2$, $\sum_{i\in\mathcal{I}}\tilde{\mathbb{E}}_{\tilde{\mu}}[w_i] \geq \alpha^2 n$.*

424

425 *Proof.* Let $u = \frac{1}{\alpha n}\tilde{\mathbb{E}}[w]$. Then, $u$ is a non-negative vector satisfying $\sum_{i\sim[n]} u_i = 1$.

426 Let $\mathsf{wt}(\mathcal{I}) = \sum_{i\in\mathcal{I}} u_i$ and $\mathsf{wt}(\mathcal{O}) = \sum_{i\notin\mathcal{I}} u_i$. Then, $\mathsf{wt}(\mathcal{I}) + \mathsf{wt}(\mathcal{O}) = 1$.

427 We will show that if $\mathsf{wt}(\mathcal{I}) < \alpha$, then there's a pseudo-distribution $\tilde{\mu}'$ that satisfies $\mathcal{A}_{w,\ell}$ and has a
428 lower value of $\|\tilde{\mathbb{E}}[w]\|_2$. This is enough to complete the proof.

429 To show this, we will "mix" $\tilde{\mu}$ with another pseudo-distribution satisfying $\mathcal{A}_{w,\ell}$. Let $\tilde{\mu}^*$ be the *actual*
430 distribution supported on single $(w,\ell)$ - the indicator $\mathbf{1}_{\mathcal{I}}$ and $\ell^*$. Thus, $\tilde{\mathbb{E}}_{\tilde{\mu}^*} w_i = 1$ iff $i \in \mathcal{I}$ and
431 $0$ otherwise. $\tilde{\mu}^*$ clearly satisfies $\mathcal{A}_{w,\ell}$. Thus, any convex combination (mixture) of $\tilde{\mu}$ and $\tilde{\mu}^*$ also
432 satisfies $\mathcal{A}_{w,\ell}$.

433 Let $\tilde{\mu}_\lambda = (1-\lambda)\tilde{\mu} + \lambda\tilde{\mu}^*$. We will show that there is a $\lambda > 0$ such that $\|\tilde{\mathbb{E}}_{\tilde{\mu}_\lambda}[w]\|_2 < \|\tilde{\mathbb{E}}[w]\|_2$.

434 We first lower bound $\|u\|_2^2$ in terms of $\mathsf{wt}(\mathcal{I})$ and $\mathsf{wt}(\mathcal{O})$. Observe that for any fixed values of $\mathsf{wt}(\mathcal{I})$
435 and $\mathsf{wt}(\mathcal{O})$, the minimum is attained by the vector $u$ that ensures $u_i = \frac{1}{\alpha n}\mathsf{wt}(\mathcal{I})$ for each $i \in \mathcal{I}$ and
436 $u_i = \frac{1}{(1-\alpha)n}\mathsf{wt}(\mathcal{O})$.

This gives $\|u\|^2 \geq \left(\frac{\mathsf{wt}(\mathcal{I})}{\alpha n}\right)^2 \alpha n + \left(\frac{1-\mathsf{wt}(\mathcal{I})}{(1-\alpha)n}\right)^2 (1-\alpha)n = \frac{1}{\alpha n} \cdot \left(\mathsf{wt}(\mathcal{I}) + (1-\mathsf{wt}(\mathcal{I}))^2 \left(\frac{\alpha}{1-\alpha}\right)\right)$.

437 Next, we compute the the $\ell_2$ norm of $u' = \frac{1}{\alpha n}\tilde{\mathbb{E}}_{\tilde{\mu}_\lambda} w$ as:

$$\|u'\|_2^2 = (1-\lambda)^2\|u\|^2 + \frac{\lambda^2}{\alpha n} + 2\lambda(1-\lambda)\frac{\mathsf{wt}(\mathcal{I})}{\alpha n}.$$

438

Thus, $\|u'\|^2 - \|u\|^2 = (-2\lambda + \lambda^2)\|u\|^2 + \frac{\lambda^2}{\alpha n} + 2\lambda(1-\lambda)\frac{\mathsf{wt}(\mathcal{I})}{\alpha n}$

$$\leq \frac{-2\lambda + \lambda^2}{\alpha n} \cdot \left(\mathsf{wt}(\mathcal{I})^2 + (1-\mathsf{wt}(\mathcal{I}))^2\frac{\alpha}{1-\alpha}\right) + \frac{\lambda^2}{\alpha n} + 2\lambda(1-\lambda)\frac{\mathsf{wt}(\mathcal{I})}{\alpha n}$$

439

Rearranging, $\|u\|^2 - \|u'\|^2 \geq \frac{\lambda}{\alpha n}\left((2-\lambda)\cdot\left(\mathsf{wt}(\mathcal{I})^2 + (1-\mathsf{wt}(\mathcal{I}))^2\left(\frac{\alpha}{1-\alpha}\right)\right) - \lambda - 2(1-\lambda)\mathsf{wt}(\mathcal{I})\right)$

$$\geq \frac{\lambda(2-\lambda)}{\alpha n}\left(\mathsf{wt}(\mathcal{I})^2 + (1-\mathsf{wt}(\mathcal{I}))^2\frac{\alpha}{1-\alpha} - \mathsf{wt}(\mathcal{I})\right)$$

440 Now, whenever $\mathsf{wt}(\mathcal{I}) < \alpha$, $\mathsf{wt}(\mathcal{I})^2 + (1-\mathsf{wt}(\mathcal{I}))^2\frac{\alpha}{1-\alpha} - \mathsf{wt}(\mathcal{I}) > 0$. Thus, we can choose a small
441 enough $\lambda > 0$ so that $\|u\|^2 - \|u'\|^2 > 0$.

442 $\qquad\qquad\qquad\qquad\qquad\qquad\qquad\qquad\qquad\qquad\qquad\qquad\qquad\qquad\qquad\qquad\qquad\qquad\qquad\qquad\square$

443 Lemma 4.3 and Lemma 4.2 immediately imply the correctness of our algorithm.

444 *Proof of Main Theorem 1.5.* First, since $D$ is $k$-certifiably $(C, \alpha\eta/4C)$-anti-concentrated,
445 Lemma 5.5 implies taking $\geq n = (kd)^{O(k)}$ samples ensures that $\mathcal{I}$ is $k$-certifiably $(C, \alpha\eta/2C)$-anti-
446 concentrated with probability at least $1 - 1/d$. Let's condition on this event in the following.

447 Let $\tilde{\mu}$ be a pseudo-distribution of degree $t$ satisfying $\mathcal{A}_{w,\ell}$ and minimizing $\|\tilde{\mathbb{E}}[w]\|_2$. Such a pseudo-
448 distribution exists as can be seen by just taking the distribution with a single-point support $w$ where
449 $w_i = 1$ iff $i \in \mathcal{I}$.

450 From Lemma 4.2, we have: $\frac{1}{|\mathcal{I}|}\sum_{i\in\mathcal{I}}\tilde{\mathbb{E}}[w_i] \cdot \|v_i - \ell^*\|_2 \leq \frac{\alpha}{2}\eta$. Let $Z = \frac{1}{\alpha n}\sum_{i\in\mathcal{I}}\tilde{\mathbb{E}}[w_i]$. By a
451 rescaling, we obtain:

$$\frac{1}{|\mathcal{I}|}\sum_{i\in\mathcal{I}}\frac{\tilde{\mathbb{E}}[w_i]}{Z}\cdot\|v_i - \ell^*\|_2 \leq \frac{1}{Z}\frac{\alpha}{2}\eta. \qquad (4.4)$$

Using Lemma 4.3, $Z \geq \alpha$. Thus,

$$\frac{1}{|\mathcal{I}|} \sum_{i \in \mathcal{I}} \frac{\tilde{\mathbb{E}}[w_i]}{Z} \cdot \|v_i - \ell^*\|_2 \leq \eta/2 \,. \tag{4.5}$$

Let $i \in [n]$ be chosen with probability $\frac{\tilde{\mathbb{E}}[w_i]}{\alpha n}$. Then, $i \in \mathcal{I}$ with probability $Z \geq \alpha$. By Markov's inequality applied to (4.5), with $\frac{1}{2}$ conditioned on $i \in \mathcal{I}$, $\|v_i - \ell^*\|_2 < \eta$. Thus, in total, with probability at least $\alpha/2$, $\|v_i - \ell^*\|_2 \leq \eta$. Thus, the with probability at least 0.99 over the draw of the random set $J$, the list constructed by the algorithm contains an $\ell$ such that $\|\ell - \ell^*\|_2 \leq \eta$.

Let us now account for the running time and sample complexity of the algorithm. The sample size for the algorithm is dictated by Lemma 5.5 and is $(kd)^{O(k)}$, which for our choice of $p$ goes as $(kd)^{O(k)}$. A pseudo-distribution satisfying $\mathcal{A}_{w,\ell}$ and minimizing $\|\tilde{\mathbb{E}}[w]\|_2$ can be found in time $n^{O(k)} = (kd)^{O(k^2)}$. The rounding procedure runs in time at most $O(nd)$. $\qquad\square$

*Remark* 4.4 (Tolerating Additive Noise). To tolerate independent additive noise, our algorithm and analysis change minimally. For an additive noise of variance $\zeta^2 \ll \alpha^2 \eta^2$ in the inliers, we modify $\mathcal{A}_{w,\ell}$ by replacing the constraint $\forall i, w_i \cdot (y_i - \langle x_i, \ell \rangle) = 0$ by $\forall i, \pm w_i \cdot (y_i - \langle x_i, \ell \rangle) \leq 4\zeta$. And $\sum_{i=1}^{n} w_i = \alpha n$ to $\sum_{i=1}^{n} w_i = (\alpha/2)n$.

This means that instead of searching for a subsample of size $\alpha n$ that has a exact solution $\ell$, we search for a subsample of size $\alpha/2n$ where there's a solution $\ell$ with an additive error of at most $2\zeta$. With additive noise of variance $\zeta^2$, it is easy to check that there's a subset of $1/2$ fraction of inliers that satisfies this property. Thus, $\mathcal{A}_{w,\ell}$ is feasible.

Our analysis remains exactly the same except for one change in the proof of Lemma 4.1. We start from a distribution that is $(C, \alpha\eta\zeta/100C)$-certifiably anti-concentrated. And instead of inferring that $p(w_i(y_i - \langle x_i, \ell \rangle)) = 1$, we use that whenever $\pm(y_i - \langle x_i, \ell \rangle) \leq 4\zeta$, $p^2((y_i - \langle x_i, \ell \rangle)) \geq 1 - 4\zeta$.

## 4.1 List-Decodable Regression for Boolean Vectors

In this section, we show algorithms for list-decodable regression when the distribution on the inliers satisfies a weaker anti-concentration condition. This allows us to handle more general inlier distributions including the product distributions on $\{\pm 1\}^d$, $[0, 1]^d$ and more generally any product domain. We however require that the unknown linear function be "Boolean", that is, all its coordinates be of equal magnitude.

We start by defining the weaker anti-concentration inequality. Observe that if $v \in \mathbb{R}^d$ satisfies $v_i^3 = \frac{1}{d} v_i$ for every $i$, then the coordinates of $v$ are in $\{0, \pm \frac{1}{\sqrt{d}}\}$.

**Definition 4.5** (Certifiable Anti-Concentration for Boolean Vectors). A $\mathbb{R}^d$ valued random variable $Y$ is $k$-*certifiably* $(C, \delta)$-anti-concentrated in *Boolean directions* if there is a univariate polynomial $p$ satisfying $p(0) = 1$ such that there is a degree $k$ sum-of-squares proof of the following two inequalities: for all $x^2 \leq \delta^2$, $(p(x) - 1)^2 \leq \delta^2$ and for all $v$ such that $v_i^3 = \frac{4}{d} v_i$ for all $i$, $\|v\|^2 \mathbb{E}_Y p(\langle Y, v \rangle)^2 \leq C\delta$.

We can now state the main result of this section.

**Theorem 4.6** (List-Decodable Regression in Boolean Directions). *For every* $\alpha, \eta$, *there's a algorithm that takes input a sample generated according to* $\mathrm{Lin}_D(\alpha, n, \ell^*)$ *in* $\mathbb{R}^d$ *for $D$ that is $k$-certifiably* $(C, \alpha\eta/10C)$-*anti-concentrated in Boolean directions and* $\ell^* \in \left\{\pm \frac{1}{\sqrt{d}}\right\}^d$ *and outputs a list $L$ of size* $O(1/\alpha)$ *such that there's an* $\ell \in L$ *satisfying* $\|\ell - \ell^*\| < \eta$ *with probability at least* 0.99 *over the draw of the sample. The algorithm requires a sample of size* $n \geq (d/\alpha\eta)^{O(\frac{1}{\alpha^2\eta^2})}$ *and runs in time* $n^{O(k)} = (d/\alpha\eta)^{O(k^2)}$.

The only difference in our algorithm and rounding is that instead of the constraint set $\mathcal{A}_{w,\ell}$, we will work with $\mathcal{B}_{w,\ell}$ that has an additional constraint $\ell_i^2 = \frac{1}{d}$ for every $i$. Our algorithm is exactly the same as Algorithm 1 replacing $\mathcal{A}_{w,\ell}$ by $\mathcal{B}_{w,\ell}$.

$$\mathcal{B}_{w,\ell} \colon \begin{cases} & \sum_{i=1}^{n} w_i = \alpha n \\ \forall i \in [n], & w_i^2 = w_i \\ \forall i \in [n], & w_i \cdot (y_i - \langle x_i, \ell \rangle) = 0 \\ \forall in \in [d], & \ell_i^2 = \dfrac{1}{d} \end{cases} \tag{4.6}$$

We will use the following fact in our proof of Theorem 4.6.

**Lemma 4.7.** *If $a, b$ satisfy $a^2 = b^2 = \frac{2}{d}$, then, $(a-b)^3 = \frac{1}{d}(a-b)$*

*Proof.* $(a-b)^3 = a^3 - b^3 - 3a^2 b + 3ab^2 = \frac{1}{d}(a - b - 3b + 3a) = \frac{4}{d}(a-b)$. □

*Proof of Theorem 4.6.* The proof remains the same as in the previous section with one additional step. First, we can obtain the analog of Lemma 4.1 with a few quick modifications to the proof. Then, Lemma 4.2 follows from modified Lemma 4.1 as in the previous section. And the proof of Lemma 4.3 remains exactly the same. We can then put the above lemmas together just as in the proof of Theorem 1.5.

We now describe the modifications to obtain the analog of Lemma 4.1. The key additional step in the proof of the analog of Lemma 4.1 which follows immediately from Lemma 4.7.

$$\left\{ \forall i \; \ell_i^2 = \frac{1}{d} \right\} \left|\frac{\ell}{4} \left\{ (\ell_i - \ell_i^*)^3 = \frac{4}{d}(\ell_i - \ell_i^*) \right\}\right.$$

This allows us to replace the usage of certifiable anti-concentration by certifiable anti-concentration for Boolean vectors and derive:

$$\left\{ \forall i \; \ell_i^2 = \frac{2}{d} \right\} \left|\frac{\ell}{4} \left\{ \frac{1}{|\mathcal{I}|} \sum_{i \in \mathcal{I}} p(\langle x_i, \ell - \ell^* \rangle)^2 \leq \frac{\alpha^2 \eta^2}{4} \right\}\right.$$

The rest of the proof of Lemma 4.1 remains the same.

□

# 5 Certifiably Anti-Concentrated Distributions

In this section, we prove certifiable anti-concentration inequalities for some basic families of distributions. We first formally state the definition of certified-anti-concentration.

**Definition 5.1** (Certifiable Anti-Concentration). A $\mathbb{R}^d$-valued zero-mean random variable $Y$ has a $(C, \delta)$-*anti-concentrated* distribution if $\Pr[|\langle Y, v \rangle| \leq \delta \sqrt{\mathbb{E}\langle Y, v \rangle^2}] \leq C\delta$.

$Y$ has a $k$-*certifiably* $(C, \delta)$-anti-concentrated distribution if there is a univariate polynomial $p$ satisfying $p(0) = 1$ such that

1. $\left\{ \langle Y, v \rangle^2 \leq \delta^2 \mathbb{E}\langle Y, v \rangle^2 \right\} \left|\frac{v}{k} \left\{ (p(\langle Y, v \rangle) - 1)^2 \leq \delta^2 \right\}\right.$.

2. $\left\{ \|v\|_2^2 \leq 1 \right\} \left|\frac{v}{k} \left\{ \|v\|_2^2 \mathbb{E}p^2(\langle Y, v \rangle) \leq C\delta \right\}\right.$.

We will say that such a polynomial $p$ "witnesses the certifiable anti-concentration of $Y$". We will use the phrases "$Y$ has a certifiably anti-concentrated distribution" and "$Y$ is a certifiably anti-concentrated random variable" interchangeably.

Before proceeding to prove certifiable anti-concentration of some important families of distributions, we observe the invariance of the definition under scaling and shifting.

**Lemma 5.2** (Scale invariance). *Let $Y$ be a $k$-certifiably $(C, \delta)$-anti-concentrated random variable. Then, so is $cY$ for any $c \neq 0$.*

*Proof.* Let $p$ be the polynomial that witnesses the certifiable anti-concentration of $Y$. Then, observe that $q(z) = p(z/c)$ satisfies the requirements of the definition for $cY$. $\qquad\square$

**Lemma 5.3** (Certified anti-concentration of gaussians)**.** *For every $0.1 > \delta > 0$, there is a $k = O\left(\frac{\log^2(1/\delta)}{\delta^2}\right)$ such that $\mathcal{N}(0, I)$ is $k$-certifiably $(2, 2\delta)$-anti-concentrated.*

*Proof.* Lemma A.1 yields that there exists an univariate even polynomial $p$ of degree $k$ as above such that for all $v$, whenever $|\langle x, v \rangle| \leq \delta, p(\langle x, v \rangle) \leq 2\delta$, and whenever $\|v\|^2 \leq 1, \mathbb{E}_{x \sim \mathcal{N}(0,I)} p(\langle x, v \rangle)^2 \leq 2\delta$. Since $p$ is even, $p(z) = \frac{1}{2}(p(z) + p(-z))$ and thus, any monomial in $p(z)$ with non-zero coefficient must be of even degree. Thus, $p(z) = q(z^2)$ for some polynomial $q$ of degree $k/2$.

The first property above for $p$ implies that whenever $z \in [0, \delta], p(z) \leq 2\delta$. By Fact 3.8, we obtain that:

$$\left\{ \langle x, v \rangle^2 \leq \delta^2 \right\} \left|\tfrac{v}{k}\right. \left\{ p(\langle x, v \rangle)^2 \leq \delta \right\}$$

Next, observe that for any $j$, $\mathbb{E}_{x \sim \mathcal{N}(0,I)} \langle x, v \rangle^{2j} = (2j)!! \cdot \|v\|_2^{2j}$. Thus, $\|v\|_2^2 \mathbb{E}_{x \sim \mathcal{N}(0,I)} p^2(\langle x, v \rangle)$ is a univariate polynomial $F$ in $\|v\|_2^2$. The second property above thus implies that $F(\|v\|_2^2) \leq C\delta$ whenever $\|v\|_2^2 \leq 1$. By another application of Fact 3.8, we obtain:

$$\left\{ \|v\|_2^2 \leq 1 \right\} \left|\tfrac{v}{k}\right. \left\{ \mathbb{E}_{x \sim \mathcal{N}(0,I)} p(\langle x, v \rangle)^2 \leq 2\delta \right\}$$

$\qquad\square$

We say that $Y$ is a *spherically symmetric* random variable over $\mathbb{R}^d$ if for every orthogonal matrix $R$, $RY$ has the same distribution as $Y$. Examples include the standard gaussian random variable and uniform (Haar) distribution on $\mathbb{S}^{d-1}$. Our argument above for the case of standard gaussian extends to any distribution that is spherically symmetric and has sufficiently light tails.

**Lemma 5.4** (Certified anti-concentration of spherically symmetric, light-tail distributions)**.** *Suppose $Y$ is a $\mathbb{R}^d$-valued, spherically symmetric random variable such that for any $k \in (0, 2)$, for all $t$ and for all $v$, $\Pr[\langle v, Y \rangle \geq t\sqrt{\mathbb{E}\langle Y, v \rangle^2}] \leq Ce^{-t^{2/k}/C}$ and for all $\eta > 0$, $\Pr_{x \sim D}[|x| < \eta\sigma] \leq C\eta$, for some absolute constant $C > 0$. Then, for $d = O\left(\frac{\log^{(4+k)/(2-k)}(1/\delta)}{\delta^{2/(2-k)}}\right)$, $Y$ is $d$-certifiably $(10C, \delta)$-anti-concentrated.*

**Lemma 5.5** (Certified anti-concentration under sampling)**.** *Let $D$ be $k$-certifiably $(C, \delta)$-anti-concentrated, subexponential and unit covariance distribution. Let $S$ be a collection of $n$ independent samples from $D$. Then, for $n \geq \Omega\left((kd\log(d))^{O(k)}\right)$, with probability at least $1 - 1/d$, the uniform distribution on $S$ is $(2C, \delta)$-anti-concentrated.*

*Proof.* Let $p$ be the degree $k$ polynomial that witnesses the certifiable anti-concentration of $D$. Let $Y$ be the random variable with distribution $D'$, the uniform distribution on $n$ i.i.d. samples from $D$. We will show that $p$ also witnesses that $k$-certifiable $(4C, \delta/2)$-anti-concentration of $Y$. To this end it is sufficient to take enough samples such that the following holds.

$$\Pr\left( \left| \mathbb{E}_D[p^2(\langle Y, v \rangle)] - \mathbb{E}_{D'}[p^2(\langle Y, v \rangle)] \right| > \mathbb{E}_D[p^2(\langle Y, v \rangle)]/2 \right) < 1/d$$

Observe that $p^2(\langle Y, v \rangle)$ may be written as $\langle c(Y)c(Y)^T, m(v)m(v)^T \rangle$ where $c(Y)$ are the coefficients of $p(\langle Y, v \rangle)$ and $m(v)$ is the vector containing monomials. The dot product above is the usual trace inner product between matrices. Now, it is sufficient to show that

$$\Pr\left( \|\mathbb{E}_{D'}c(Y)c(Y)^T - \mathbb{E}_D c(Y)c(Y)^T\|_F^2 > \|\mathbb{E}_D c(Y)c(Y)^T\|_F^2/4 \right) < 1/d$$

Since $p$ was a univariate polynomial of degree $k$ in $d$ dimensional variables, there are at most $d^{2k}$ entries in total, and each entry is at most a degree $2k$ polynomial of subexponential random variables in $d$ variables. Using standard concentration results for polynomials of subexponential random variables (for instance Theorem 1.2 from [27] and the references therein). We see that each entry satisfies

$$\Pr\left( |\mathbb{E}_D c(Y)_i c(Y)_j - \mathbb{E}_{D'} c(Y)_i c(Y)_j| > \epsilon \right) \leq \exp\left( -\Omega\left( \frac{n\epsilon}{\mathbb{E}(c(Y)_i c(Y)_j)^2} \right)^{1/2k} \right)$$

An application of a union bound, squaring the term inside and replacing $\epsilon^2$ by $\mathbb{E}(c(Y)_i c(Y)_j)^2/4$ gives us

$$\Pr\left(\sum_{i,j=1}^{d^{2k}} \left(\mathbb{E}_D c(Y)_i c(Y)_j - \mathbb{E}_{D'} c(Y)_i c(Y)_j\right)^2 > \|\mathbb{E}c(Y)c(Y)^T\|_F^2/4\right) \le d^{2k}\exp\left(-\Omega\left(\frac{n}{d^{O(k)}}\right)^{1/2k}\right)$$

Hence, setting $n = O((kd\log(d))^{O(k)})$ ensures that with probability at least $1 - 1/d$, the distribution $D'$ is $(2C, \delta)$-anti-concentrated.

$\square$

We say that a $d \times d$ matrix $A$ is $C'$-well-conditioned if all singular values of $A$ are within a factor of $C'$ of each other.

**Lemma 5.6** (Certified anti-concentration under linear transformations). *Let $Y$ be $k$-certifiably $(C, \delta)$-anti-concentrated random variable over $\mathbb{R}^d$. Let $A$ be any $C'$-well-conditioned linear transformation. Then, $AY$ is $k$-certifiably $(C, C'^2\delta)$-anti-concentrated.*

*Proof.* Let $\|A\|$ be the largest singular value of $A$. Let $p$ be a polynomial that witnesses the certifiable anti-concentration of $Y$. Let $q(z) = p(z/\|A\|)$. We will prove that $q$ witnesses the $k$-certifiable $(C, C'^2\delta)$-anti-concentration of $AY$.

Towards this, observe that:

$$\left\{\langle Y, v\rangle^2 \le \delta^2 \mathbb{E}\langle Y, v\rangle^2\right\} \left|\frac{v}{2}\right. \left\{\langle AY, v\rangle^2 \le \delta^2 \mathbb{E}\langle AY, v\rangle^2\right\}.$$

$$\left\{\langle Y, (A^T v)/\|A\|\rangle^2 \le \delta^2 \mathbb{E}\langle Y, (A^T v)/\|A\|\rangle^2\right\} \left|\frac{v}{k}\right. \left\{(p(\langle Y, (A^T v)/\|A\|\rangle) - 1)^2 \le \delta^2\right\},$$

this is the same as

$$\left\{\langle AY, v\rangle^2 \le \delta^2 \mathbb{E}\langle AY, v\rangle^2\right\} \left|\frac{v}{k}\right. \left\{(q(\langle AY, v\rangle) - 1)^2 \le \delta^2\right\}.$$

Where $q = p(x/\|A\|)$. Now, for $w = (A^T v)/\|A\|$ and any unit vector $v$,

$$\left\{\|w\|_2^2 \le 1\right\} \left|\frac{v}{k}\right. \left\{\|A^T v\|_2^2/\|A\|_2^2 \mathbb{E}p^2(\langle AY, v\rangle/\|A\|) \le C\delta\right\},$$

Thus,

$$\left\{\|A^T v\|_2^2 \le \|A\|^2\right\} \left|\frac{v}{k}\right. \left\{\|A^T v\|_2^2 \mathbb{E}q^2(\langle AY, v\rangle) \le C\|A\|_2^2\delta\right\}.$$

However,

$$\left\{\|v\|_2^2 \le 1\right\} \left|\frac{v}{2}\right. \left\{\|A^T v\|_2^2 \le \|A\|^2\right\},$$

and thus,

$$\left\{\|v\|_2^2 \le 1\right\} \left|\frac{v}{k}\right. \left\{\|v\|_2^2 \mathbb{E}q^2(\langle AY, v\rangle) \le CC'^2\delta\right\}.$$

$\square$

**Lemma 5.7** (Certifiable Anti-Concentration in Boolean Directions). *Fix $C > 0$. Let $Y$ be a $\mathbb{R}^d$ valued* product *random variable satisfying:*

    *1. **Identical Coordinates**: $Y_i$ are identically distributed for every $1 \le i \le d$.*

    *2. **Anti-Concentration** For every $v \in \left\{0, \pm\frac{1}{\sqrt{d}}\right\}^d$, $\Pr[|\langle Y, v\rangle| \le \delta\sqrt{\mathbb{E}\langle Y, v\rangle^2}] \le C\delta$.*

    *3. **Light tails** For every $v \in \mathbb{S}^{d-1}$, $\Pr[|\langle Y, v\rangle| > t\sqrt{\mathbb{E}\langle Y, v\rangle^2}] \le \exp(-t^2/C)$.*

*Then, $Y$ is $k$-certifiably $(C, \delta)$-anti-concentrated for $k = O\left(\frac{\log^2(1/\delta)}{\delta^2}\right)$.*

*Proof.* We use the $p$ from Lemma A.1. To see that $p$ witnesses the anti-concentration of $Y$, once again observe that Lemma A.1 applies to give us a real life proof of the required statements. We now exhibit a sum of squares proof. Observe that every monomial of even degree $2k$ for any $k \in \mathbb{N}$, $\mathbb{E}_{Y \sim D}\langle Y, v \rangle^{2k}$ is a *symmetric* polynomial in $v$ with non-zero coefficients only on even-degree monomials in $v$. This follows by noting that the coordinates of $D$ are independent and identically distributed and $x^2$ is an even function. It is a fact that all symmetric polynomials in $v$ can be expressed as polynomials in the "power-sum" polynomials $\|v\|_{2i}^{2i}$ for $i \leq 2t$. However, since $v_i^2 \in \left\{0, \frac{1}{d}\right\}$ for $i \geq 1$, $\|v\|_{2i}^{2i} = \frac{1}{d^{i-1}}\|v\|_2^2$. Hence a polynomial in $\|v\|_{2i}^{2i}$ is also a univariate polynomial in $\|v\|_2^2$. Since these are polynomial inequalities, they are also sum-of-squares proofs of these inequalities.

The observation above implies $\|v\|_2^2 \mathbb{E}_Y p(\langle Y, v \rangle)^2 = \|v\|_2^2 \cdot F(\|v\|_2^2)$ for some degree $k$ univariate polynomial $F$. Since Since $F$ is a univariate polynomial and $\|v\|_2^2 \leq 1$ is an "interval constraint" by applying Fact 3.8, we get: $\left|\frac{\|v\|_2^2}{2t}\right| \left\{\|v\|_2^2 F(\|v\|_2^2) \leq C\delta\right\}$. Recalling the fact that $\|v\|_2^2 \mathbb{E}_Y p(\langle Y, v \rangle)^2 = \|v\|_2^2 \cdot F(\|v\|_2^2)$, this completes the proof. $\square$

# 6 Information-Theoretic Lower Bounds for List-Decodable Regression

In this section, we show that list-decodable regression on $\mathrm{Lin}_D(\alpha, \ell^*)$ information-theoretically requires that $D$ satisfy $\alpha$-anti-concentration: $\Pr_{x \sim D}[\langle x, v \rangle = 0] < \alpha$ for any non-zero $v$.

**Theorem 6.1** (Main Lower Bound). *For every $q$, there is a distribution $D$ on $\mathbb{R}^d$ satisfying $\Pr_{x \sim D}[\langle x, v \rangle = 0] \leq \frac{1}{q}$ such that there's no $\frac{1}{2q}$-approximate list-decodable regression algorithm for $\mathrm{Lin}_D(\frac{1}{q}, \ell^*)$ that can output a list of size $< d$.*

*Remark* 6.2 (Impossibility of Mixed Linear Regression on the Hypercube). Our construction for the case of $q = 2$ actually shows the impossibility of the well-studied and potentially easier problem of noiseless *mixed linear regression* on the uniform distribution on $\{0, 1\}^n$. This is because $\mathcal{R}_i$ is, by construction, obtained by using one of $e_i$ or $\mathbf{1} - e_i$ to label each example point with equal probability.

Theorem 6.1 is tight in a precise way. In Proposition 2.4, we proved that whenever $D$ satisfies $\Pr_{x \sim D}[\langle x, v \rangle = 0] < \frac{1}{q}$, there is an (inefficient) algorithm for *exact* list-decodable regression algorithm for $\mathrm{Lin}_D(\frac{1}{q}, \ell^*)$. Note that our lower bound holds even in the setting where there is no additive noise in the inliers.

Somewhat surprisingly, our lower bound holds for extremely natural and well-studied distributions - uniform distribution on $\{0, 1\}^n$ and more generally, uniform distribution on $\{0, 1, \ldots, q-1\}^d = [q]^d$ for any $q$. We can easily determine a tight bound on the anti-concentration of both these distributions.

**Lemma 6.3.** *For any non-zero $v \in \mathbb{R}^d$, $\Pr_{x \sim \{0,1\}^n}\langle x, v \rangle = 0 \leq \frac{1}{2}$ and $\Pr_{x \sim [q]^d}[\langle x, v \rangle = 0] \leq \frac{1}{q}$.*

Note that this is tight for any $v = e_i$, the vector with 1 in the $i$th coordinates and 0s in all others.

*Proof.* Fix any $v$. Without loss of generality, assume that all coordinates of $v$ are non-zero. If not, we can simply work with the uniform distribution on the sub-hypercube corresponding to the non-zero coordinates of $v$.

Let $S \subseteq \{0, 1\}^n$ ($[q]^d$, respectively) be the set of all $x \in \{0, 1\}^n$ ($[q]^d$, respectively) such that $\langle x, v \rangle = 0$. Then, observe that for any $x \in S$, and any $i$, $x^{(i)}$ obtained by flipping the $i$th bit (changing the $i$th coordinate to any other value) of $x$ cannot be in $S$. Thus, $S$ is an independent set in the graph on $\{0, 1\}^n$ (in $[q]^d$, respectively) with edges between pairs of points with hamming distance 1.

It is a standard fact [56] that the maximum independent set in the $d$-hypercube is of size exactly $2^{d-1}$ and in the $q$-ary Hamming graph $[q]^d$ is of size $q^{d-1}$. Thus, $\Pr_{x \sim \{0,1\}^d}[\langle x, v \rangle = 0] \leq \frac{1}{2}$ and $\Pr_{x \sim [q]^d}[\langle x, v \rangle = 0] \leq \frac{1}{q}$.

$\square$

To prove our lower bound, we give a family of $d$ distributions on labeled linear equations, $\mathcal{R}_i$ for $1 \leq i \leq d$ that satisfy the following:

1. The examples in each are chosen from uniform distribution on $[q]^d$,

2. $\frac{1}{q}$ fraction of the samples are labeled by $e_i$ in $\mathcal{R}_i$, and,

3. for any $i, j$, $\mathcal{R}_i$ and $\mathcal{R}_j$ are statistically indistinguishable.

Thus, given samples from $\mathcal{R}_i$, any $\frac{1}{2q}$-approximate list-decoding algorithm must produce a list of size at least $d$.

Our construction and analysis of $\mathcal{R}_i$ is simple and exactly the same in both the cases. However it is somewhat easier to understand for the case of the hypercube ($q = 2$). The following simple observation is the key to our construction.

**Lemma 6.4.** *For $1 \leq i \leq d$, let $\mathcal{R}_i$ be the distribution on linear equations induced by the following sampling method: Sample $x \sim \{0,1\}^d$, choose $a \sim \{0,1\}$ uniformly at random and output: $(x, \langle x, (1-a)e_i \rangle)$. Then, $\mathcal{R}_i = \mathcal{R}_j$ for any $i, j \leq d$.*

*Proof.* The proof follows by observing that $\mathcal{R}_i$ when viewed as a distribution on $\mathbb{R}^{d+1}$ is same as the uniform distribution on $\{0,1\}^{d+1}$ and thus independent of $i$. $\qquad\square$

The argument immediately generalizes to $[q]^d$ and yields:

**Lemma 6.5.** *For $1 \leq i \leq d$, let $\mathcal{R}_i$ be the distribution on linear equations induced by the following sampling method: Sample $x \sim [q]^d$, choose $a \sim \{0,1\}$ uniformly at random and output: $(x, (\langle x, e_i \rangle + a) \mod q)$. Then, $\mathcal{R}_i = \mathcal{R}_j$ for any $i, j \leq d$.*

In this case, we interpret the $1/q$ fraction of the samples where $a = 0$ as the inliers. Observe that these are labeled by a single linear function $e_i$ in any $\mathcal{R}_i$. Thus, they form a valid model in $\text{Lin}_D(\alpha, \ell^*)$ for $\alpha = 1/q$.

Since the linear functions defined by $e_i$ on $[q]^d$, when normalized to have unit norm, have a pairwise Euclidean distance of at least $1/q$, we immediately obtain a proof of Theorem 6.1.

# A  Polynomial Approximation for Core-Indicator

The main result of this section is a low-degree polynomial approximator for the function $\mathbf{1}(|x| < \delta)$ with respect to all distributions that have asymptotically lighter-than-exponential tails.

**Lemma A.1.** *Let $D$ be a distribution on $\mathbb{R}$ with mean $0$, variance $\sigma^2 \leq 1$ and satisfying:*

1. ***Anti-Concentration:*** *For all $\eta > 0$, $\Pr_{x \sim D}[|x| < \eta\sigma] \leq C\eta$, and,*

2. ***Tail bound:*** $\Pr[|x| \geq t\sigma] \leq e^{-\frac{t^{2/k}}{C}}$ *for $k < 2$ and all $t$,*

*for some $C > 1$. Then, for any $\delta > 0$, there is a $d = O\left(\frac{\log^{(4+k)/(2-k)}(1/\delta)}{\delta^{2/(2-k)}}\right) = \tilde{O}\left(\frac{1}{\delta^{2/(2-k)}}\right)$ and an even polynomial $q(x)$ of degree $d$ such that $q(0) = 1$, $q(x) = 1 \pm \delta$ for all $|x| \leq \delta$ and $\sigma^2 \cdot \mathbb{E}_{x \sim D}\left[q^2(x)\right] \leq 10C\delta$.*

Before proceeding to the proof, we note that the bounds on the degree above are tight up to poly logarithmic factors for the gaussian distribution.

**Lemma A.2.** *For every polynomial $p$ of degree $d$ such that $p(0) = 1$, $\mathbb{E}_{x \sim \mathcal{N}(0,1)}[p^2(x)] = \Omega\left(\frac{1}{\sqrt{d}}\right)$. Further, there is a polynomial $p_*$ of degree $d$ such that $p_*(0) = 1$ and $\mathbb{E}_{x \sim \mathcal{N}(0,1)}p_*^2(x) = \Theta\left(\frac{1}{\sqrt{d}}\right)$.*

Our construction of the polynomial is based on standard techniques in approximation theory for constructing polynomial approximators for continuous functions over an interval. Most relevant for us are various works of Eremenko and Yuditskii [24, 25, 23] and Diakonikolas, Gopalan, Jaiswal, Servedio and Viola [13] on such constructions for the sign function on the interval $[-1, a] \cup [a, 1]$ for $a > 0$. We point the reader to the excellent survey of this beautiful line of work by Lubinsky [43].

676 **Fact A.3** (Theorem 3.5 in [13])**.** *Let $0 < \eta < 0.1$, then there exist constants $C, c$ such that for*

$$a := \eta^2/C \log(1/\eta) \text{ and } K = 4c \log(1/\eta)/a + 2 < O(\log^2(1/\eta)/\eta^2)$$

677 *there is a polynomial $p(t)$ of degree $K$ satisfying*

678     *1. $p(t) > \text{sign}(t) > -p(-t)$ for all $t \in \mathbb{R}$.*

679     *2. $p(t) \in [\text{sign}(t), \text{sign}(t) + \eta]$ for $t \in [-1/2, -2a] \cup [0, 1/2]$.*

680     *3. $p(t) \in [-1, 1 + \eta]$ for $t \in (-2a, 0)$*

681     *4. $|p(t)| \le 2 \cdot (4t)^K$ for all $t > \frac{1}{2}$.*

682 We will also rely on the following elementary integral estimate.

**Lemma A.4** (Tail Integral)**.**

$$\int_{[L,\infty]} \exp\left(-\frac{x^{2/k}}{C}\right) x^{2d} dx < \exp\left(-\frac{L^{2/k}}{C}\right)\left((L)^{4d} + (16kd)^{kd}\right).$$

683 *Proof.* We first prove the claim for $k = 1$. Let $y = x - L$. The, $\int_L^\infty e^{-x^2} x^{2d} dx = \int_0^\infty e^{-(y+L)^2}(y +$
684 $L)^{2d} dy$. We now use that $y^2 + L^2 \le (y + L)^2$ for all $y \ge 0$ and $(y + L)^{2d} \le 2^{2d}(y^{2d} + L^{2d})$ to
685 upper bound the integral above by: $e^{-L^2} L^{2d} + 2^{2d} e^{-L^2} \int_0^\infty e^{-y^2} y^{2d}$. Using $\int_0^\infty e^{-y^2} y^{2d} < (4d)^d$
686 gives a bound of $e^{-L^2}(L^{2d} + (8d)^d)$.

687 For larger $k$, we substitute $y = x^{1/k}$ and write the integral in question as $\int_{L^{1/k}}^\infty e^{-y^2} y^{2kd-(k-1)} dy$.
688 Applying the calculation from the above special case, this integral is upper bounded by: $e^{-L^{2/k}}(L^{4d} +$
689 $(16kd)^{kd})$.        □

690 *Proof of Lemma A.1.* Let $p(x)$ be the degree $d < O\left(\frac{L \log^2(1/\delta)}{\delta}\right)$ polynomial from Fact A.3. We
691 then construct a polynomial $q(x)$ that will be close to 0 in the range $[\delta, L]$ and $[-L, -\delta]$ and close to
692 1 in the range $[-\delta, \delta]$. Our polynomial $q$ is obtained by shifting and appropriately scaling two copies
693 of $p$.

$$q(x) = \frac{p\left(a + \frac{x}{4L}\right) + p\left(-(a + \frac{x}{4L})\right) - 1}{p(a) + p(-a) - 1}$$

694 Then, $q(0) = 1$. It further satisfies:

695     1. $q(x) \in [0, C\sqrt{\delta/L}]$ for $x \in [\delta, L] \cup [-L, \delta]$.

696     2. $q(x) \in [1 - C\sqrt{\delta/L}, 1 + \sqrt{\delta/L}]$ for $x \in [-\delta, \delta]$.

697     3. $q(x) \in [0, 1 + \sqrt{\delta/L}]$ for $x \in [-3\delta, -\delta] \cup [\delta, 3\delta]$.

698     4. $|q(x)| < 4 \cdot (4x)^t$ for $|x| > L$

699 We now prove the bound the $\mathbb{E}p^2$. We do this by providing upper bounds on the contributions to
700 $\sigma^2 \cdot \mathbb{E}_{x \sim \mathcal{D}}\left[q^2(\sigma x)\right]$ from the disjoint sets with different guarantees below. Since we are going to
701 evaluate $q(\sigma x)$ the intervals will be scaled by $\sigma$.

702 The contributions from the regions $\frac{1}{\sigma}[\delta, L]$ and $\frac{1}{\sigma}[-\delta, \delta]$ can be naively upper bounded by the
703 maximum value that the polynomial can take here times the probability of landing in these regions.
704 The first of these contributes $\sigma \cdot \frac{\delta}{L} \cdot (L - \delta) \le \delta$, and using anticoncentration, the second region
705 contributes $\left(1 + \sqrt{\frac{\delta}{L}}\right)^2 \cdot 2C\delta \le 4C\delta$. The region $\frac{1}{\sigma}[\delta, 3\delta]$ can be bounded similarly to get an upper
706 bound of $2\left(1 + \sqrt{\frac{\delta}{L}}\right)^2 \sigma^2 \delta \le 4\delta$. To finish, we use Lemma A.4 to upper bound the contribution to

707 $\mathbb{E}p^2$ from the tail:

$$\sigma^2 C' \int_{\frac{1}{\sigma}[L,\infty]} q^2(\sigma x) \exp\left(-\frac{x^{2/k}}{C}\right) dx \lesssim \sigma^{2+d} 4^d \exp\left(-\frac{1}{C}\cdot\left(\frac{L}{\sigma}\right)^{2/k}\right)\left((L/\sigma)^{4d} + (16kd)^{kd}\right)$$

$$\lesssim \exp\left(2d + 4d\log\left(\frac{L}{\sigma}\right) - \frac{1}{C}\cdot\left(\frac{L}{\sigma}\right)^{2/k} + kd\log(16kd)\right)$$

708 We choose $L$ satisfying $10d\log(d) + 4d\log(\frac{L}{\sigma}) - \frac{1}{C}\cdot(\frac{L}{\sigma})^{2/k} < 2\log(1/\delta)$.

709 Since $d = O\left(\frac{L\log^2(1/\delta)}{\delta}\right)$, $k < 2$, and $\sigma < 1$ we can now choose $L = \left(\frac{C100\log^3(1/\delta)}{\delta}\right)^{k/(2-k)}$ to

710 satisfy the inequality above and to get $d \lesssim \frac{\log^{2+3k/(2-k)}(1/\delta)}{\delta^{1+k/(2-k)}}$. When $k = 1$ we get $d = \tilde{O}(1/\delta^2)$.

711 Since $\sigma < 1$ in all the above calculations, we get our result by re-scaling $\delta$.

712 $\qquad\qquad\qquad\qquad\qquad\qquad\qquad\qquad\qquad\qquad\qquad\qquad\qquad\qquad\qquad\qquad\qquad\qquad\qquad\quad\square$

713 We now complete the proof of Lemma A.2.

714 *Proof of Lemma A.2.* Any polynomial $p$ of degree $d$ can be written as $p(x) = \sum_{i=1}^d \alpha_i h_i(x)$ where

715 $h_i$ denote the hermite polynomials of degree $i$, satisfying $\mathbb{E}_{x\sim\mathcal{N}(0,1)}h_i = 0$ and $\mathbb{E}_{x\sim N(0,1)}[h_i^2(x)] =$

716 1. Since $p(0) = 1$, using Cauchy-Schwartz inequality, we obtain:

$$\mathbb{E}_{x\sim N(0,1)}[p^2(x)]\cdot\sum_{i=1}^d h_i^2(0) = \left(\sum_{i=1}^d \alpha_i^2\right)\cdot\left(\sum_{i=1}^d h_i^2(0)\right) \geq \left(\sum_{i=1}^d \alpha_i h_i(0)\right)^2 \geq 1$$

717 Further, observe that for the polynomial $p_*(x) = \frac{1}{\sum_i h_i^2(0)}\sum_i h_i(0)h_i(x)$, the above inequality is

718 tight. Using that $h_{2i}(0) = \frac{(2i-1)!!}{\sqrt{(2i)!}}$ and $h_i(0) = 0$ if $i$ is odd, (see, for e.g., [55]), we have:

$$\mathbb{E}_{x\sim N(0,1)}[p^2(x)] \geq \mathbb{E}_{x\sim\mathcal{N}(0,1)}p_*^2(x) = \left(\sum_{i=1}^d h_i^2(0)\right)^{-1} = \left(\sum_{i=1}^{d/2}\left(\frac{(2i-1)!!}{\sqrt{(2i)!}}\right)^2\right)^{-1}$$

$$= \left(\sum_{i=1}^{d/2}\frac{(2i)!}{2^{2i}i!^2}\right)^{-1} = \left(\sum_{i=1}^{d/2}\binom{2i}{i}\cdot\frac{1}{2^{2i}}\right)^{-1} = \Theta\left(\sum_{i=1}^{d/2}\frac{1}{\sqrt{i}}\right)^{-1} = \Theta\left(\sqrt{d}\right)^{-1}.$$

719 $\qquad\qquad\qquad\qquad\qquad\qquad\qquad\qquad\qquad\qquad\qquad\qquad\qquad\qquad\qquad\qquad\qquad\qquad\qquad\quad\square$

# B Brute-force search can generate a $\exp(d)$ size list

720

721 In the following, we write $e_i$ to denote the vector with $1$ in the $i$th coordinate and $0$s in all others.

722 **Proposition B.1.** *There exists a distribution $D$ on $\mathbb{R}^d$ and a model $\mathrm{Lin}_D(\alpha, \ell^*)$ such that for every*

723 $\alpha < 1/2$, *with probability at least $1 - 1/d$ over the draw of a $n$-size sample $\mathcal{S}$ from $\mathrm{Lin}_D(\alpha, \ell^*)$, there*

724 *exists a collection $\mathrm{Sol} \subseteq \{S \subseteq \mathcal{S} \mid |S| = \alpha n\}$ of size $\exp(d)$ and unit length vectors $\ell_S$ for every*

725 $S \in \mathrm{Sol}$ *such that $\ell_S$ satisfies all equations in $S$ and for every $S \neq S' \in \mathrm{Sol}$, $\|\ell_S - \ell_{S'}\|_2 \geq 0.1$.*

726 *Proof.* Let $D$ be the uniform distribution on $e_1, e_2, \ldots, e_d \in \mathbb{R}^d$. Let $\ell^* := \vec{1}/\sqrt{d}$ be the all-ones

727 vector in $\mathbb{R}^d$ scaled by $1/\sqrt{d}$ and let $d$ samples be drawn from the uncorrupted distribution. These give

728 us our inliers, $\mathcal{I} = \{(x_i, y_i)\}_{i=1}^{\alpha n}$. For the outliers, choose the following multiset $\mathcal{O} := 1/\alpha - 1$ copies

729 of $\{(e_i, j) \mid i \in [d], j \in \{\pm 1/\sqrt{d}\}\}$. This is a sample set of size $2d/\alpha$. Any $a \in \{\pm 1/\sqrt{d}\}^d$ is a valid

730 candidate for a solution for this data. This is because for any such $a$, $\mathcal{I}_a := \{(e_i, a_i) \mid i \in [d]\} \subset S$

731 satisfies the following

732       1. $\mathcal{I}_a \subset S, |\mathcal{I}_a| = d = \frac{\alpha}{2}|S|$ and

733       2. for any $(x, y) \in \mathcal{I}_a$, $y = \langle x, a \rangle$.

The Gilbert–Varshamov bound from coding theory now tells us that there are at least $\Omega(\exp(\Omega(d)))$ $\{0,1\}$ vectors in $d$ dimensions that pairwise have a hamming distance of $0.1 \cdot d$. This transfers to the set $\{\pm 1/\sqrt{d}\}$ to give us that there are $\Omega(\exp(\Omega(d)))$ vectors in $\{\pm 1/\sqrt{d}\}$ that are pairwise $0.1$ apart in 2-norm.

$\square$

## Footnotes

[1]There's a long line of work on robust regression algorithms (see for e.g. [8, 33]) that can tolerate corruptions only in the *labels*. We are interested in algorithms robust against corruptions in both examples and labels.

[5]The choice of norm is not important here because the factor $2^{-n^\ell}$ swamps the effects of choosing another norm.

[6]Here, we assume that the bitcomplexity of the constraints in $\mathcal{A}$ is $(n + m)^{O(1)}$.