[Reviews · NeurIPS 2019]

Reviewer 1



Post-rebuttal response: I read the authors' response and don't have any further comments. ------------- ORIGINALITY =========== Linear regression in the presence of noise is a fundamental problem in machine learning. This paper considers the model of "robust statistics" where an alpha fraction of the training data comes from the ground truth distribution while the rest are corrupted arbitrarily (i.e. adversarially). Traditionally, research has been on the setting where alpha is large, so that the parameters of the true distribution are information-theoretically identifiable. However, recent focus has been on the small alpha setting. Here, the parameters of the true distribution cannot be uniquely identified even with infinitely many samples. However, it may be possible to output a small list of parameters which is guaranteed to contain the ground truth. Previous work in this regime has focused on mean estimation. Here, the authors give the first efficient list-decodable learning algorithms for linear regression. In my opinion, the work is quite original, both because of the foundational nature of the problem and also because of the new technical contributions, as I discuss below. QUALITY & CLARITY ================== The submission is well-written and sound. I verified at a high level most of the proofs in the supplementary material and was convinced. Also, the authors do a good job of motivating and giving intuition for their algorithm and the anti-concentration condition they require of the input distribution. SIGNIFICANCE ============ The techniques introduced in this paper should be influential in the future for analyzing other list-decoding learning algorithms. The work relies on the "identifiablity to algorithms" paradigm that has been very successful recently in designing a variety of robust learning algorithms. The key idea is to first design a non-efficient algorithm that involves searching over distributions, then to relax the search over distributions to "pseudo-distributions" (which is in poly time), and then to devise a "rounding" scheme that allows use of the pseudo-distribution in the rest of the algorithm. In this work, the target distribution mu is over subsets of the input examples, so that (1) each set S in supp(mu) is of size >= alpha*n, (2) each set S in supp(mu) is consistent with a linear hypothesis, and (3) for any i, the marginal probability of i being included in a random subset S from mu is approximately the same. Given such a mu, the authors give a nice and simple list-decoding learning algorithm, assuming that the samples come from an anti-concentrated distribution. Next, the search over distributions is relaxed to a search over pseudo-distributions. Condition 3 is relaxed to an optimization in terms of "pseudo-expectations". The whole optimization problem is now in poly time via the ellipsoid algorithm. Finally, the issue of rounding, which is where this paper shows its originality. The non-efficient algorithm samples from mu O(1/alpha) times and returns the list of consistent linear functions. But with a pseudo-distribution, it doesn't make sense to obtain independent samples. So, what they do is to let each i output a linear function v_i that is the average according to the pseudo-distribution mu~ of all the linear functions corresponding to the sets S in supp(mu) that contain i. They show that assuming "certifiable anticoncentration", if the i'th sample is uncorrupted, then with probability 1/2, v_i is close to the true linear function l*. But if we sample i with probability proportional to the marginal probability of the pseudo-distribution to i, then i is in the uncorrupted set with probability at least alpha. So, with probability at least alpha/2, v_i is going to be close to l*, and we are done. The authors state, and I agree, that this method of "rounding by votes" could be used in other settings as well. The paper also shows that the gaussian distribution and more general spherically symmetric distributions are certifiably anti-concentrated. This uses an application of approximation theory to construct a polynomial approximator for the indicator function of an interval. Also, if the true linear function is boolean, then they show that the anti-concentration assumption can be relaxed so as to allow product distributions (such as uniform). Finally, they also show that anti-concentration is necessary for identifiability even for noiseless regression with examples from the uniform distribution on {0,1}^d, and even for the potentially simpler problem of mixed linear regression.

Reviewer 2



This paper considers the problem of linear regression when the majority of the data may be adversarial "outliers", i.e., when there is a small set of interest for which we would like estimates of the regression parameters. The paper makes the key assumptions that (1) the inlier distribution is sufficiently anti-concentrated (as many natural distributions are) and (2) the set has at least constant (alpha) measure. Then they propose a polynomial time algorithm (exponential in alpha) that is guaranteed to produce a list of ~ 1/alpha candidate parameters that includes an accurate estimate of the regression parameters for the unknown alpha-dense subset. I note that while the work of Charikar-Steinhardt-Valiant proposed a general convex optimization algorithm for learning in this setting, the guarantee they obtain for their algorithm is sufficiently weak that a trivial 0 solution would essentially always meet their guarantee, and indeed the algorithm does not estimate regression parameters in practice. On the other hand, work by Kothari-Steinhardt-Steurer from last year on some simpler problems in the list-decodable learning model considered using an assumption that Poincare inequalities were known to hold in the sum-of-squares framework; the assertion of an anticoncentration inequality is along these lines, but I do not think that this specific assumption was considered previously, and indeed even the formulation as a sum-of-squares statement could be considered a contribution. But, there is at least one work that solves list-decodable regression in the alpha << 1/2 regime, albeit with a very different dependence on the parameters: in AISTATS this year, Hainline et al. obtained a guarantee with a polynomial dependence on 1/alpha, but an exponential dependence on the number of nonzero coefficients in the regression parameters; their list size is also polynomial, and not the (near-)optimal ~1/alpha list size obtained here. Indeed, note that Remark 1.7 (on the exponential dependence on 1/alpha) only applies when the list size is small, and the lower bound in Theorem 6.1 showing the necessity of anti-concentration only applies when the list size is at most the dimension, but I do not believe that using a polynomial-size list trivializes the problem. The techniques of Hainline et al. are very different, using a RANSAC-like algorithm, that indeed also does not rely on an anti-concentration property for the inlier distribution. I view these works as complementary, especially as Hainline et al. scaled exponentially with the number of coefficients, so while the present work is still a significant contribution, this prior work does reduce the novelty of the guarantee slightly. In any case, the guarantees are interesting and the paper is overall very clear. The interesting part is that anti-concentration guarantees identifiability for linear regression, which is very clearly explained. A small note to the authors: v is not specified in Definition 1.2 (although 2.1 corrects this omission), so it doesn't make sense as stated. ======== Regarding the comparison to Hainline et al.: you are correct that the problem that is the focus of that work is not the same. But, nevertheless, a connection to the list-decodable learning model is discussed in that paper. Their analysis shows that for an arbitrary set of inliers, a good approximation (under the desired l_p loss) to the optimal regression coefficients on that set is found on some iteration of the main loop. So, if one simply outputs the list of solutions obtained across iterations of the loop, this algorithm solves the list-decodable sparse linear regression problem. You are correct that the loop is a search over tuples of points, thus suffering an exponential dependence on the number of nonzero regression coefficients. I certainly agree that it is interesting that the algorithm proposed in your work has a polynomial dependence on the dimension, and that the techniques are different and interesting. But, again, note that the "brute-force" algorithm avoids the exponential dependence on 1/alpha your algorithm suffers. This is perhaps most interesting in contrast to the lower bounds you discuss; note that such an enumeration of tuples of points cannot be captured in the standard SQ model, either. It thus illustrates the weakness/limitations of these lower bounds.

Reviewer 3



OVERVIEW. The algorithm is obtained using the sum-of-squares (SoS) approach, where the constraints of the algorithm are expressed as polynomial inequalities, and the algorithm finds a "pseudo-distribution" satisfying those inequalities using convex programming. The solution is then extracted from the pseudo-distribution by a randomized rounding procedure based on voting. The total running time in this approach is exponential in the degree of the polynomials. One of the main novel ideas introduced in the paper is how to reason about anticoncentration (i.e., upper bounding the probability that a random sample falls into a small neighborhood) in the SoS framework. The approach here is to approximate the indicator function of the neighborhood by a low-degree polynomial. The overall submission is somewhat technical, but less so than a typical paper using the SoS approach, which is a plus. In any case, the main ideas are well explained in the first 7 pages. EVALUATION. The result is quite interesting. There are relatively few algorithms for robust estimation where the fraction of outliers alpha is arbitrarily close to 1 (notably for mean estimation), so a new algorithm in this regime is somewhat of a rarity. Furthermore, obtaining the result required several novel ideas. At the same time, given the high runtime bound of the algorithm, the result seems only of theoretical interest (and there is no experimental evaluation showing otherwise). So the potential audience for this paper at NeurIPS is probably small.

[Author Response · NeurIPS 2019]

We thank the reviewers for their comments.

Regarding the exponential dependence on $1/\alpha$, two of our colleagues have independently informed us that they can
prove SQ lower bounds showing this dependence is necessary. We have encouraged them to write up their proofs as
soon as possible.

Regarding the Hainline et al. paper, we thank the reviewer for bringing this paper to our attention and will cite it in
the related work section. That said, we are a bit puzzled by the direct comparison to our work. In Hainline et al, they
consider the problem of *conditional* linear regression, where the goal is to find a linear function with small square loss
*conditioned* on a subset of training points whose 'indices' satisfy some constant-width $k$-DNF formula.

In our paper, there are simply inliers (points from the true distribution) and outliers, and we give an algorithm that
outputs a list of linear functions, one of which has small error with respect to the inliers. We show that this algorithm
succeeds if and only if the inliers are drawn from an anti-concentrated distribution. We do not place any 'computational
constraints' on the training set (such as satisfiability by DNF formulas).

Finally, the Hainline et al. paper uses a brute force search subroutine over all subsets of the training set (of size '$r$') and
thus obtain consequences for regression where the coefficient vectors have a constant number of non-zero entries. The
point of our work is to avoid brute force search by developing new techniques relying on the sum-of-squares method.

[Meta-Review · NeurIPS 2019]

This paper studies the challenging problem of doing linear regression in the setting where an overwhelming fraction (1-alpha) of the examples are adversarially corrupted. It extends recent work on using the Sum-of-Squares hierarchy for robust estimation. The main contribution is realizing that anti-concentration (and being able to certify anti-concentration) is the key. The algorithm has a high running time (d^(1/alpha^8)) but given the challenging nature of the problem, the reviewers felt that the fact that the problem can be solved in polynomial time for any fixed alpha > 0 is surprising and an important contribution.